# Active Acquisition for Multimodal Temporal Data: A Challenging Decision-Making Task

**Jannik Kossen**[1][*][†]   **Cătălina Cangea**[2][†]   **Eszter Vértes**[2]   **Andrew Jaegle**[2]
**Viorica Patraucean**[2]   **Ira Ktena**[2]   **Nenad Tomasev**[2]   **Danielle Belgrave**[2]

[1] *OATML, Department of Computer Science, University of Oxford*
[2] *Google DeepMind*

**Reviewed on OpenReview:** *https://openreview.net/forum?id=Gbu1bHQhEL*

## Abstract

We introduce a challenging decision-making task that we call *active acquisition for multimodal temporal data* (A2MT). In many real-world scenarios, input features are not readily available at test time and must instead be acquired at significant cost. With A2MT, we aim to learn agents that actively select which modalities of an input to acquire, trading off acquisition cost and predictive performance. A2MT extends a previous task called *active feature acquisition* to temporal decision making about high-dimensional inputs. We propose a method based on the Perceiver IO architecture to address A2MT in practice. Our agents are able to solve a novel synthetic scenario requiring practically relevant cross-modal reasoning skills. On two large-scale, real-world datasets, Kinetics-700 and AudioSet, our agents successfully learn cost-reactive acquisition behavior. However, an ablation reveals they are unable to learn adaptive acquisition strategies, emphasizing the difficulty of the task even for state-of-the-art models. Applications of A2MT may be impactful in domains like medicine, robotics, or finance, where modalities differ in acquisition cost and informativeness.

## 1   Introduction

In making a clinical diagnosis, the medical professional must carefully choose a specific set of tests to diagnose the patient quickly and correctly. It is of crucial importance to choose the right test at the right time, and tests should only be performed when useful, as they may otherwise cause unnecessary patient discomfort or financial expense. Recently, large-scale datasets of medical treatment records have become available (Hyland et al., 2020; Johnson et al., 2020). They may potentially facilitate improvements in medical domain knowledge and patient care, for example by allowing us to learn which tests to perform when. While prior work has demonstrated that machine learning can be used to inform complex diagnoses from simple measurements, see, for example, Liotta et al. (2003); Diaz-Pinto et al. (2022), treatment records present a significant modelling challenge as they contain temporally sparse observations from high-dimensional modalities, e.g. X-Rays, MRIs, blood tests, or genetic data.

Prior work in *active feature acquisition* (AFA) (Greiner et al., 2002; Melville et al., 2004; Ling et al., 2004; Zubek et al., 2004; Sheng & Ling, 2006; Saar-Tschansky et al., 2009) has similarly considered the cost of feature acquisition at test time on a per-datum basis: given a test input for which all features are missing initially, which features should one acquire to best trade off predictive performance and the cost of feature acquisition? This is different to active learning (Settles, 2010), which minimizes the number of *label* acquisitions needed for model *training.* Concurrent methods in AFA often use a Bayesian experimental design approach (Lindley, 1956; Chaloner & Verdinelli, 1995; Sebastiani & Wynn, 2000), acquiring features that maximise the expected information gain with respect to the prediction (Ma et al., 2018; Li & Oliva,

---

[*]Work done while interning at Google DeepMind.
[†]Correspondence to jannik.kossen@cs.ox.ac.uk and ccangea@deepmind.com.

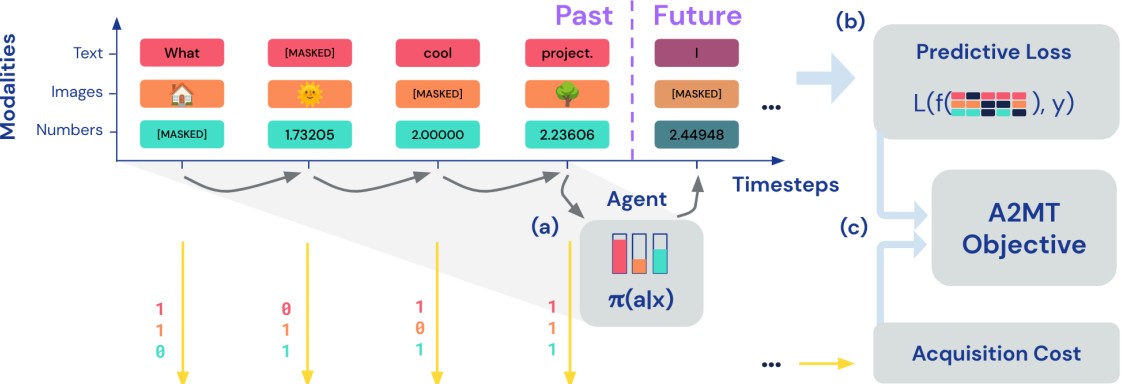

Figure 1: In many practical applications, features are not available a priori at test time and have to be acquired at a real-world cost to allow for the prediction of an associated label. In *Active Acquisition for Multimodal Temporal Data*, we aim to learn agents that efficiently acquire for multimodal temporal inputs: (a) at each timestep, the agent decides which modalities of the input it acquires, paying a per-modality acquisition cost; (b) then, a separate model predicts given the sparse sequence of observations; (c) lastly, the agent gets rewarded for low prediction loss and small acquisition cost.

2021; Lewis et al., 2021). Alternatively, AFA can be phrased as a reinforcement learning task where agents optimize the trade-off objective directly (Shim et al., 2018; Kachuee et al., 2019; Zannone et al., 2019; Janisch et al., 2019; 2020; Yin et al., 2020). Notably, prior work in AFA assumes static data: although acquisitions are sequential, feature values do not evolve along a temporal dimension. Furthermore, the features themselves usually correspond to low-dimensional observations, i.e. single values in a tabular dataset.

In this work, we propose *active acquisition for multimodal temporal data* (A2MT). Taking the above medical setup as motivation, we extend the familiar setting of AFA in two key ways: (1) We assume that inputs are sequences that evolve temporally. Our agents will need to learn not only which features to acquire, but also *when* to acquire them. (2) We no longer assume that inputs are unimodal and low-dimensional. Instead, we assume that each input comprises a collection of high-dimensional modalities, and that acquisitions are made for entire modalities at each timestep. With these extensions, A2MT generalizes AFA and reduces the gap to practical applications that are often both temporal and multimodal. A2MT can also find application outside the medical domain (cf. §6).

To study A2MT in a controlled environment, we propose a set of synthetic scenarios of increasing difficulty that are temporal *and* multimodal, both key requirements for A2MT. Further, we propose to study A2MT on audio-visual datasets, concretely AudioSet (Gemmeke et al., 2017) and Kinetics-700 2020 (Smaira et al., 2020). These provide a challenging testbed for A2MT and avoid some of the complications of working with medical data. We propose a method based on Perceiver IO (Jaegle et al., 2021)—a modality-agnostic architecture that can be applied directly to a large variety of real-world inputs—and we explore different reinforcement learning techniques to train the agent. Our method is able to solve a subset of the synthetic tasks we propose and provides reasonable performance on the real-world datasets. However, further investigation reveals that our method can ultimately be outperformed by a non-adaptive ablation, highlighting the difficulty of the A2MT task and, consequently, opportunities for future work.

In summary, our contributions are:

- We introduce the *active acquisition for multimodal temporal data* (A2MT) scenario (§2).
- We suggest both synthetic and real-world datasets that motivate A2MT and allow for convenient benchmarking of methods (§3).
- We propose a novel method based on Perceiver IO to tackle A2MT (§4), provide a thorough empirical study (§5), and discuss key areas of improvement for future work (§7).

## 2 Active Acquisition for Multimodal Temporal Data

With A2MT (fig. 1), we study strategies for cost-sensitive acquisition when data is both multimodal and temporal. We train an agent policy $\pi$ that makes a binary acquisition decision per modality and timestep. After a fixed number of timesteps, a model $f$ makes a sequence-level prediction given the sparsely acquired observations. The agent then observes a reward that consists of two terms: (1) The agent gets rewarded in proportion to the negative predictive loss given the observations. (2) Acquisitions come at a (modality-specific) cost and the agent is penalized for each acquisition made. Term (1) compels the agent to acquire often, as additional observations should improve the quality of the predictions of $f$. However, this increases the penalty incurred from term (2). The agent therefore needs to trade off acquisition cost and prediction reward, learning which modalities in the input are worth acquiring and when it is worth acquiring them.

To make the most of a limited acquisition budget, we aim to learn agents with individualized acquisition strategies that adapt to the sequences at hand. For example, agents may learn to make use of interactions between modalities: a past observation in one modality (e.g. a suspicious value in a blood test) may lead to the acquisition of another modality (e.g. a more specialised test), achieving both accuracy and cost-efficiency. A2MT requires models that can successfully accommodate sparse, multimodal, and temporal data. This is challenging for state-of-the-art models, even without the additional complexities of active selection required for successful A2MT.

In A2MT (Alg. 1), we assume access to a fully observed training set of input-output pairs $((\boldsymbol{x}_1, y_1), \ldots, (\boldsymbol{x}_N, y_N))$. For classification tasks, for example, $y_i$ are labels, $y_i \in (1, \ldots, C)$. Each input $\boldsymbol{x}_i$ is a sequence of observations $\boldsymbol{x}_i = (\boldsymbol{x}_{i,1}, \ldots, \boldsymbol{x}_{i,T})$. At each timestep $t$, the observation $\boldsymbol{x}_{i,t}$ decomposes into $M$ modalities $\boldsymbol{x}_{i,t} = (\boldsymbol{x}_{i,t,1}, \ldots, \boldsymbol{x}_{i,t,M})$; each modality may be high-dimensional, $\boldsymbol{x}_{i,t,m} \in \mathbb{R}^{d_m}$. For example, $\boldsymbol{x}_{i,t,m}$ could be a single frame in a video where we collapse the image height $H$, width $W$, and color channels $C$ into a single axis with dimensionality $d_m = H \cdot W \cdot C$. We focus on a single sample and drop the leading axis, $\boldsymbol{x} = \boldsymbol{x}_i$, to avoid notational clutter. We use colons to indicate 'slices' of inputs along a particular axis, e.g. $\boldsymbol{x}_{1:t-1} = (\boldsymbol{x}_1, \ldots, \boldsymbol{x}_{t-1})$.

At each timestep $t \in (1, \ldots, T)$, we obtain a set of acquisition decisions across modalities, or actions, $\boldsymbol{a}_t = (a_{t,1}, \ldots, a_{t,M})$, by sampling from the agent policy, $\boldsymbol{a}_t \sim \pi(\cdot | \tilde{\boldsymbol{x}}_{1:t-1}, a_{1:t-1}; \boldsymbol{\theta})$. Here, $a_{t,m} \in (0, 1)$ is a binary indicator of whether modality $m$ was acquired at time $t$. We write $\tilde{\boldsymbol{x}}$ instead of $\boldsymbol{x}$ to highlight the fact that the inputs may contain missing entries, and $\boldsymbol{\theta}$ are the trainable agent parameters. Here, $\pi$ gives the joint probability over all possible acquisition decisions at that timestep, including extremes such as acquiring all or none of the modalities, i.e. $\pi(\boldsymbol{a}_t | \tilde{\boldsymbol{x}}_{1:t-1}, a_{1:t-1}; \boldsymbol{\theta}) \in [0, 1]^M$. At each timestep $t$ and for each modality $m$, we acquire $\boldsymbol{x}_{m,t}$ only if $a_{m,t} = 1$. We summarize the set of all actions across timesteps as $\boldsymbol{a} = (\boldsymbol{a}_1, \ldots, \boldsymbol{a}_T)$.

---

**Algorithm 1** A2MT

**Inputs:** Test input $\boldsymbol{x}$, agent $\pi$, model $f$.

1: **for** $t = 1$ to $T$ **do**
2:      Sample $a_t \sim \pi(\cdot | \tilde{\boldsymbol{x}}_{1:t-1}, \boldsymbol{a}_{1:t-1}; \boldsymbol{\theta})$.
3:      **for** $m = 1$ to $M$ **do**
4:          **if** $a_{t,m} = 1$ **then**
5:              Acquire: $\tilde{\boldsymbol{x}}_{t,m} \leftarrow \boldsymbol{x}_{t,m}$.
6:          **else**
7:              Do not acquire: $\tilde{\boldsymbol{x}}_{t,m} \leftarrow \emptyset$.
8:          **end if**
9:      **end for**
10: **end for**
11: Return prediction $f(\tilde{\boldsymbol{x}}_{1:T})$.

---

Lastly, we assume access to a model $f$ which makes a prediction $\hat{y}$ given an input $\tilde{\boldsymbol{x}}$. After completing the acquisition process for a given test sample, we use this model to predict $f(\tilde{\boldsymbol{x}}_{1:T}) = \hat{y}$. Consequently, we require that $f$ can predict for multimodal inputs with features missing arbitrarily across time and modalities.

We train agents to maximize the following reward:

$$R = \mathbb{E}\left[-C(\boldsymbol{a}) - \mathcal{L}(f(\tilde{\boldsymbol{x}}_{1:T}), y)\right], \quad \text{where} \quad C(\boldsymbol{a}) = \sum_{t=1}^{T} \sum_{m=1}^{M} c_m a_{t,m}. \tag{1}$$

Here, the expectation is over the data $(\boldsymbol{x}, y)$ and actions $\boldsymbol{a}$ from the policy; $C(\boldsymbol{a})$ gives the total cost of acquisition along the sequence; $c_m$ is a modality-specific cost, and $\mathcal{L}(f(\tilde{\boldsymbol{x}}_{1:T}), y)$ is log likelihood loss. Equation (1) summarizes the problem of A2MT: trading off acquisition costs against predictive performance

for multimodal and temporal inputs. In §4, we detail how we optimize this objective using reinforcement learning. We refer to §D for a discussion of A2MT in terms of Markov Decision Processes.

## 3 Datasets for A2MT

While the medical domain acts as important practical *motivation* for A2MT, medical datasets, such as Hyland et al. (2020); Johnson et al. (2020), usually require significant domain expertise. And while the eventual application of A2MT methods in this domain is crucial, we believe that a mix of synthetic and real *non-medical* data can serve as a widely accessible testbed for the machine learning community to develop robust methodology. We therefore introduce datasets suitable for developing A2MT methods.

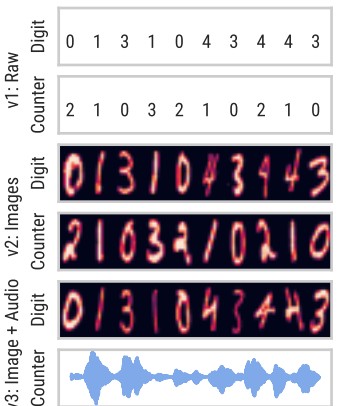

Figure 2: The synthetic scenarios allow for sparse acquisition while keeping perfect accuracy. This requires agents capable of cross-modal reasoning. (Label is 9 in the above.)

### 3.1 Synthetic Datasets

We introduce three scenarios with varying levels of difficulty and a clear optimal acquisition strategy, designed to test the cross-modal reasoning capabilities of the agents.

Concretely, we create a dataset (cf. algorithm B.1) of fixed-length sequences with two modalities: `counter` and `digit`. The `digit` modality is a sequence of random numbers of length $T$, e.g. `0131043443`. For the `counter` modality, we draw random 'starting values' uniformly from the set of valid numbers. For each starting value, we create a sequence counting down to 0 from that starting value, and then concatenate all countdown sequences. I.e. when drawing the starting values 2, 3, and 2, we generate the sequences `210`, `3210`, and `210`, and the final counter modality is their concatenation, `2103210210`. We cut sequences to length $T$. The label attached to each sequence is the sum of the `digit` modality at all timesteps where the `counter` modality is zero, e.g. `3+3+3=9` in this case.

To increase the complexity of this task, we can replace the sequence of raw numbers with a sequence of images, for example replacing raw numbers with matching MNIST digits (LeCun, 1998) for both modalities. We also consider replacing the numbers of the `counter` modality with *audio* sequences from the spoken digit dataset (Jackson, 2017). Figure 2 shows these synthetic dataset variants. Further, one can increase task complexity by adjusting the sequence length and the number of unique symbols for the modalities.

To solve the synthetic task, agents need to reason about interactions between modalities; a key feature of our practical motivation for A2MT. Each modality in the synthetic scenario offers an ideal strategy to save acquisition cost without sacrificing predictive accuracy. (1) The agent can learn to acquire the `digit` modality only if the `counter` modality is zero, because only then is a `digit` relevant for the label. (2) The agent can further learn to *skip* observations in the `counter` modality, because of the regular pattern they follow.

### 3.2 Audio-Visual Datasets

Additionally, we propose to apply large-scale audio-visual classification datasets, such as AudioSet (Gemmeke et al., 2017) or Kinetics-700-2020 (Smaira et al., 2020), to the A2MT setting. These datasets are multimodal and temporal, each input being a sequence of sound and images. We divide each input video into a set of temporally aligned images and audio segments. Compared to the synthetic scenarios, these datasets offer the complexity and noise of real-world data, ideally allowing A2MT agents to optimize the trade-off objective in interesting ways.

In fig. 3, we illustrate the variety in inputs for the AudioSet dataset: (a) shows an example where both modalities are informative towards the label, in (b) the change in the image modality could be a signal for the agent to revisit the audio modality, (c) shows an example where the image modality is seemingly unrelated to the sound-based label, and in (d) all image frames are identical.

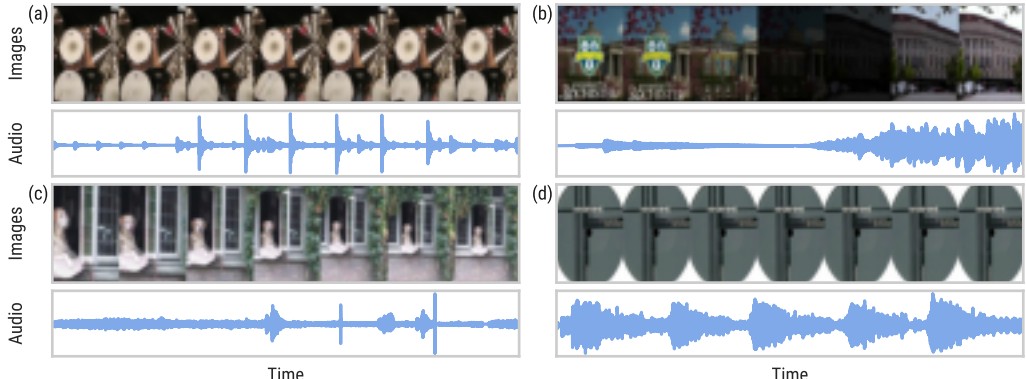

Figure 3: Four example sequences from the AudioSet training set. The labels associated with these inputs are (a) Music, (b) Music, (c) Speech, and (d) Electronic music. The audio signal is often more informative of the label than the images for AudioSet. Inputs are downsampled in the above visualization.

We observe empirically that model predictions degrade consistently if we mask out parts of the input at evaluation time (cf. §5). Further, we find that models trained on inputs which span longer durations (increasing the temporal stride to keep the input size constant) perform better. These results suggest there is temporal variety in these datasets and that additional observations improve model predictions. This supports the idea that A2MT-style selection of the *right* inputs is possible.

## 4 Perceiver IO for A2MT

We use Perceiver IO (Jaegle et al., 2021) for both the predictive model $f$ and the agent $\pi$. Perceiver IO is a suitable architecture for A2MT as it is modality-agnostic and scales well to large input sizes. Further, Perceiver IO gracefully handles missing data: mask tokens are used to signal where features are missing in the input, cf. Jaegle et al. (2021). For the model $f$, we make no changes to the standard Perceiver IO architecture. For the agent $\pi$, we condition on previous actions by appending them to the *decoder* input of the Perceiver. We propose two variants of our approach: Inputs for the synthetic datasets tend to be somewhat smaller, and we can afford a computationally more involved routine. In contrast, for the real world datasets, we use a computationally leaner setup that allows for sequential application of the Perceiver and training of the agent at scale. We give full details in §C.

### 4.1 Variant 1: Small Data Regime

For the synthetic scenarios, the agent $\pi$ and the model $f$ are two fully separate Perceiver IO models that do not share any parameters. They are trained jointly: the agent acquires the input data features for the classifier. For policy training, we can make use of the straight-through Gumbel-Softmax estimator (Jang et al., 2017) due to the simple unmasking effect of actions in A2MT. The Gumbel trick (Jang et al., 2017; Maddison et al., 2017) allows for backpropagation through the discrete action variables and is generally considered a low-variance estimator in comparison to alternatives such as REINFORCE (Williams, 1992).

Concretely, joint training directly follows algorithm 1 for each input in a batch of training samples: we iteratively apply the agent $\pi$, unmasking modalities as given by the sampled actions $a$. We then predict with the model $f$ given the partially observed input sequence $\tilde{\boldsymbol{x}}_{1:T}$. We can compute a Monte Carlo sample of the reward, eq. (1), given the observed prediction loss $\mathcal{L}(f(\tilde{\boldsymbol{x}}_{1:T}), y)$ and cost of acquisition $C(\mathbf{a})$, where $\mathcal{L}$ is log-likelihood loss. Due to the Gumbel parameterization of the actions, we can directly apply gradient-based optimization to the agent parameters $\boldsymbol{\theta}$. We simultaneously train the model $f$ by minimizing the loss $\mathcal{L}$ in the objective with respect to the parameters of $f$.

### 4.2 Variant 2: Large Data Regime

We apply the Perceiver-based agent *repeatedly* during A2MT, i.e. up to $T = 25$ times in the scenarios that we study. This increases computational cost, particularly for gradient computation during training. To

save compute for the large scale inputs of real-world datasets, we share Perceiver IO encoders between the predictive model $f$ and agent $\pi$; unlike in the small data regime, $f$ and $\pi$ are no longer fully separate models. We further pre-train the encoder and the decoder of $f$. In other words, in the large data regime, the classifier $f$ is no longer trained on inputs acquired by the policy $\pi$. Lastly, when training the agent decoder parameters, we fix both the (shared) encoder parameters as well as the decoder parameters of $f$.

To ensure that the classifier—and, in particular, the shared encoder—is suitable for inputs encountered later during agent training, we propose to use a particular masking setup for the inputs during pre-training of the classifier. Concretely, for each input, **(M1)** we drop modalities at timestep $t$ by sampling masks with fixed per-modality probability, $a_{m,t} \sim \mathrm{Bernoulli}(p_m)$; **(M2)** we randomly draw $t_{\max} \sim \mathrm{Unif}(0, T)$ and mask out *all* inputs at $t > t_{\max}$, i.e. $a_{m,t} = 0 \,\forall m \,\forall t > t_{\max}$; **(M3)** we randomly drop entire modalities from the inputs with fixed per-modality rates $a_{m,t} = d_m \,\forall t$, with $d_m \sim \mathrm{Bernoulli}(p_m^{(d)})$. The masking mechanisms **(M1-M3)** are applied only during pre-training, and we use the same fixed values for $p_m$ and $d_m$ in all experiments, see §C.4 for further details.

This masking procedure exposes the encoder to sparse input distributions during pre-training that are equivalent to those created by a randomly acting agent. This leads to versatile encoder representations that are useful during agent training. In addition to helping agent training, we observe that the masking procedures affect the test set performance of model predictions positively, and we observe the highest performance when all methods (**M1**–**M3**) are applied simultaneously (cf. §5). This fits with similar results on how Transformer-based architectures benefit from masking at training time (Devlin et al., 2018; Dosovitskiy et al., 2021; Feichtenhofer et al., 2022).

Following related work in active feature acquisition, such as Li & Oliva (2021), we additionally condition the agent directly on model predictions $f(\tilde{\boldsymbol{x}}_{1:t})$ at each timestep $t$. Concretely, we concatenate both predicted class probabilities, $p_f(y|\tilde{\boldsymbol{x}}_{1:t})$, as well as associated predictive entropies, $\mathbb{E}_{p_f(y|\tilde{\boldsymbol{x}}_{1:t})}[-\log p_f(y|\tilde{\boldsymbol{x}}_{1:t})]$, to the latent representation of the Perceiver IO agent. While this incurs additional computational cost from predicting with the model at each timestep, we believe the additional information will be useful in informing agent behavior. For example, this allows the agent to be aware of the uncertainty attached to model predictions at each timestep. This way, the agent could, for example, learn to stop acquiring when model predictions are sufficiently confident.

Inspired by reward shaping (Ng et al., 1999; Li & Oliva, 2021), we optionally add an intermediate reward term to the objective eq. (1), $I = -\alpha \sum_{t=1}^{T} \left( \mathcal{L}(f(\tilde{x}_{1:t}), y) - \gamma \mathcal{L}(f(\tilde{x}_{1:t-1}), y) \right)$, where $\alpha$ is a hyperparameter, and $\gamma$ is the discount factor. This term directly incentivizes the agent to decrease the predictive loss of $f$, which helps with credit assignment, as it is otherwise hard for the agent to learn which particular acquisition actually led to loss reduction.

The Gumbel-Softmax estimator becomes prohibitively expensive in the large-scale scenario, as it requires full backpropagation through repeated application of both $f$ and $\pi$. We therefore rely on the advantage actor critic (A2C) (Sutton & Barto, 2018) policy gradient method, which approximates gradients of eq. (1) with respect to the agent parameters via Monte Carlo samples of the score function estimator. A2C uses an additional baseline model that reduces variance during optimization, and we implement this baseline as a separate Perceiver IO decoder head.

## 5 Experiments

We next give experimental results on synthetic and real data and refer to §C for further details.

### 5.1 Synthetic Scenario

We begin by exploring the performance of our small-scale variant on the synthetic datasets. We use sequences of length $T = 10$ and three distinct values $(0, 1, 2)$ per modality such that the input shape is $(T, \mathbb{R}^{d_m}) = (10, 3)$ per modality when using a one-hot encoding for the values.

```
                | Input Sequence 1      | Input Sequence 2      | Input Sequence 3      |
----------------+-----------------------+-----------------------+-----------------------
Digit           | [2 0 0 2 0 0 2 2 0 1] | [2 1 2 2 0 1 0 1 1 1] | [1 2 1 1 2 1 0 1 0 1] |
Actions Digit   | [0 1 0 0 1 0 0 1 0 0] | [0 1 1 0 0 1 0 0 1 0] | [0 1 0 0 1 0 0 1 0 0] |
Counter         | [1 0 2 1 0 2 1 0 2 1] | [2 1 0 2 1 0 2 1 0 2] | [1 0 2 1 0 2 1 0 2 1] |
Actions Counter | [1 1 1 1 0 1 0 1 1 0] | [0 1 1 1 0 0 1 0 1 0] | [1 1 1 1 0 1 0 1 1 0] |
Label           | True 2   / Pred.: 2   | True 4   / Pred.: 4   | True 5   / Pred.: 5   |
```

Figure 4: Acquisition behavior of the Perceiver IO agent on a simple synthetic scenario. 'Digit' and 'Counter' give ground truth values for the `Counter` and `Digit` input modalities. 'Actions Digit' and 'Actions Counter' mark when the agent did (`1`) or did not acquire (`0`) for each of the modalities. The agent successfully learns a sparse acquisition strategy: it (almost always) acquires the `Digit` modality only if the `Counter` modality is 0, and further learns to skip some acquisitions in the `Counter` modality.

For raw digits as inputs our agent learns to acquire an average of 46.2% of the `digit` modality, 68.8% of the `counter` modality, and achieves an accuracy of 92.4% on the test set. Clearly, the agent learns a selective acquisition procedure for both modalities without sacrificing predictive accuracy. We further compute the optimal acquisition rates for the synthetic scenario as 37.2% for the `digit` and 39.3% for the `counter`

Table 1: Results on the synthetic task.

| Metric | Agent | Oracle |
|---|---|---|
| Label Prediction Accuracy | 92.4% | 100.0% |
| `Digit` Acquisition Rate | 46.2% | 37.2% |
| `Counter` Acquisition Rate | 68.8% | 39.3% |

modality. These highlight that, in particular for the `counter` modality, there is a gap relative to optimal behavior for our agent, cf. table 1. In §A, we further compare the agent's acquisition strategy to optimal behavior. We display examples of learned agent behavior on individual test set inputs in fig. 4. For the digit modality, the agent mostly follows the ideal strategy and acquires only whenever the counter is zero. For the counter modality, the agent learns to skip acquisitions at some of the non-informative timesteps. In fig. A.1, we display training dynamics: the agent initially shows high acquisition rates for both modalities, which quickly leads to increases in model predictive accuracy. Then, the agent learns to discard irrelevant parts of the input; this happens more quickly for the digit than the counter modality.

Figure A.2 shows training curves for early results on the image and image/audio versions of the synthetic scenario. The agents overfit to the training set and their behavior does not generalize. We suspect that our method struggles to learn acquisition strategies and representations simultaneously.

## 5.2 Audio-Visual Datasets

Next, we investigate the performance of the large scale variant of our approach on the AudioSet and Kinetics datasets. We split the training set of each dataset into a subset used for model pre-training and a subset used exclusively for agent training, taking up 80% and 20% of the original training set respectively. Note again that model and agent share the encoder, which is learned during model pre-training, and then fixed during agent training. For both datasets, each input sample consists of audio-visual input with 250 frames of images and a raw audio signal spanning 10 seconds. For images, we take every 10th frame as input, obtaining a total of 25 input frames. We pre-process the audio signal to mel spectrograms, and then divide the signal into 25 segments. This leads to an input shape of $(25, 40 \cdot 128)$ for the audio modality, and $(25, 200 \cdot 200 \cdot 3)$ for the image modality, where we collapse additional dimensions beyond time into the last axis as described in §2. We first discuss insights from pre-training before moving on to results for A2MT agents.

### 5.2.1 Model Pretraining

**Masking Variants.** Table 2 gives results of different pre-training variants for Perceiver IO on AudioSet. We observe that masking significantly improves mean average precision (mAP; higher is better) on the AudioSet validation set, going from 0.178 mAP without any masking to 0.344 mAP for the complete masking setup **(M1–M3)**

Table 2: Masking at training time as proposed by **(M1-M3)** improves performance on the (unmasked, fully observed) AudioSet test set.

| Variant | mAP |
|---|---|
| No Masking | 0.178 |
| **(M1)** Random Masking | 0.230 |
| + **(M2)** Max. Timestep | 0.262 |
| + **(M3)** Modality Dropping | 0.344 |
| + Conv. Downsampling | 0.370 |

Table 3: Impact of dropping entire modalities at evaluation time on the AudioSet and Kinetics test sets. For both datasets, performance is best when no modalities are dropped (fully observed). For AudioSet, audio is more informative than images, as 'audio only' performance trumps 'image only'; for Kinetics, images are more informative.

| Variant | AudioSet (mAP) | Kinetics (top-1) |
|---|---|---|
| Fully Observed | 0.370 | 0.396 |
| Audio Only | 0.302 | 0.087 |
| Image Only | 0.137 | 0.305 |

Table 4: Gradually masking out the more informative modality (audio for AudioSet, images for Kinetics) at evaluation time. Performance degrades as the 'mask rate', the fraction of randomly selected timesteps $t$ at which we mask inputs, increases. The weak modality (images for AudioSet, audio for Kinetics) is not masked here.

| Mask Rate | AudioSet (mAP) | Kinetics (top-1) |
|---|---|---|
| 0.0 | 0.370 | 0.396 |
| 0.3 | 0.369 | 0.397 |
| 0.8 | 0.331 | 0.365 |
| 0.9 | 0.292 | 0.316 |
| 1.0 | 0.137 | 0.087 |

(cf. §4.2). We observe that each of the masking procedures **(M1-M3)** progressively improves evaluation metrics. See §C.4 for details on masking rates. We further add a single strided convolutional pre-processing layer per modality, which further improves performance. Note that, here, we evaluate the model on *fully observed data*. For A2MT we are particularly interested in the predictive performance of $f$ for sparse data.

**Sparsity at Evaluation Time.** In order for A2MT agent training to be successful, the model $f$ needs to be capable of extracting information from *sparse* inputs. Table 3 reports model performance when dropping entire modalities during evaluation. We observe that the model relies on both modalities for prediction, as the performance on fully observed data is best. As expected, the audio modality is more informative for AudioSet and images are stronger for Kinetics. Table 4 shows how predictions degrade when increasing the masking rate across time for the stronger modality. Predictions are best on fully observed data and deteriorate significantly at 90% missing inputs. Evidently, the model learns to make use of additional data in the input at low masking rates while also predicting reasonably for sparse inputs.

In summary, the proposed masking routines **(M1-M3)** help learn models which use all modalities in the input and degrade as inputs become increasingly sparse. These results suggest that our pre-trained Perceiver IO models are good candidates for use in A2MT, which we investigate next.

### 5.2.2 Agent Training

**Agents React to Cost.** We follow the setup described in §4.2 and train agents using A2C without intermediate rewards. We set a nonzero acquisition cost only for the more informative modality of each dataset. Table 5 shows agent behavior for acquisition costs on the audio modality of the AudioSet dataset and table 6 gives results for image cost on Kinetics. We observe that our agents generally react to increased acquisition costs by decreasing the number of acquisitions, learning how many acquisitions are 'worth it' for a given cost. Further, we find that agents hold on to a single acquisition (1/25 inputs frames, i.e. acquisition rate 0.04) even at relatively high costs. This makes sense, as performance deteriorates drastically when all of the informative modality is dropped. Conversely, agents readily learn to not acquire large portions of the informative input modality, as we have observed this has only a small effect on predictive scores. For the Kinetics dataset, the agents discover that an acquisition rate of about 0.2 optimizes the objective across a variety of low to medium costs. In §A, we discuss results when imposing costs on the weak modality.

**Random Ablations.** We are ultimately interested in learning agents which display *adaptive* behavior that meaningfully adjusts to the information in each input. Therefore, we compare the performance of our agents against two ablations that we call 'random-rate' and 'random-1hot'. For both, we first compute the average acquisition rate of the agent per modality, i.e. the average fraction of timesteps at which it acquires. For the 'random-rate' ablation, we acquire modalities with a fixed Bernoulli probability per timestep equal to the average acquisition rate of the agent. Additionally, we construct the 'random-1hot' ablation by acquiring at a fixed number of timesteps per modality that are equidistantly spread across the sequence. The number of acquisitions is chosen such that we match the average number of agent acquisitions per modality. (Usually

Table 5: The agent acquisition rate appropriately reduces for the audio modality on the **AudioSet** test set as acquisition costs increase; the agent acquisition rate is the average fraction of timesteps at which the agent acquires. We impose costs only on the audio modality and compare against two ablations: (a) Random-Rate, which acquires at each timestep with a fixed probability matching the agent acquisition rate, and (b) Random-1Hot, which acquires at a fixed set of equidistant timesteps matching the acquisition rate of the agent. The agent outperforms the rate-matched ablation at high costs, but does not improve over the discrete 1-hot ablation. See the main text for further discussion. Standard deviations are 5-fold repetitions on the test set.

| Cost per Audio Acquisition | $1 \times 10^{-6}$ | $1 \times 10^{-5}$ | $1 \times 10^{-4}$ | $2.5 \times 10^{-4}$ | $1 \times 10^{-3}$ |
|---|---|---|---|---|---|
| Agent Acquisition Rate | 0.98 | 0.83 | 0.23 | 0.11 | 0.04 |
| Agent (mAP) | $0.3724 \pm 0.0010$ | $0.3697 \pm 0.0025$ | $0.3359 \pm 0.0117$ | $0.3116 \pm 0.0108$ | $0.2580 \pm 0.0010$ |
| Random-Rate (mAP) | $0.3728 \pm 0.0005$ | $0.3719 \pm 0.0007$ | $0.3365 \pm 0.0016$ | $0.2946 \pm 0.0014$ | $0.2295 \pm 0.0009$ |
| Random-1Hot (mAP) | $0.3732 \pm 0.0001$ | $0.3731 \pm 0.0002$ | $0.3472 \pm 0.0003$ | $0.3165 \pm 0.0006$ | $0.2660 \pm 0.0004$ |

Table 6: The agent acquisition rates generally reduce as costs increase for the image modality of the **Kinetics** test set; the agent acquisition rate is the average fraction of timesteps at which the agent acquires. We do not impose any costs for acquisitions of the less-informative audio modality. We report standard deviations over 5-fold repeated application on the test set. See main text or table 5 for details.

| Cost per Image Acquisition | $1 \times 10^{-3}$ | $5 \times 10^{-3}$ | $1 \times 10^{-2}$ | $1 \times 10^{-1}$ | $5 \times 10^{-1}$ |
|---|---|---|---|---|---|
| Agent Acquisition Rate | 0.19 | 0.22 | 0.21 | 0.09 | 0.04 |
| Agent (top1) | $0.3370 \pm 0.0009$ | $0.3336 \pm 0.0034$ | $0.3326 \pm 0.0016$ | $0.3309 \pm 0.0011$ | $0.2722 \pm 0.0005$ |
| Random-Rate (top1) | $0.2804 \pm 0.0019$ | $0.2856 \pm 0.0019$ | $0.2830 \pm 0.0006$ | $0.2378 \pm 0.0012$ | $0.1721 \pm 0.0014$ |
| Random-1Hot (top1) | $0.3038 \pm 0.0003$ | $0.3067 \pm 0.0003$ | $0.3063 \pm 0.0002$ | $0.2957 \pm 0.0007$ | $0.2089 \pm 0.0002$ |

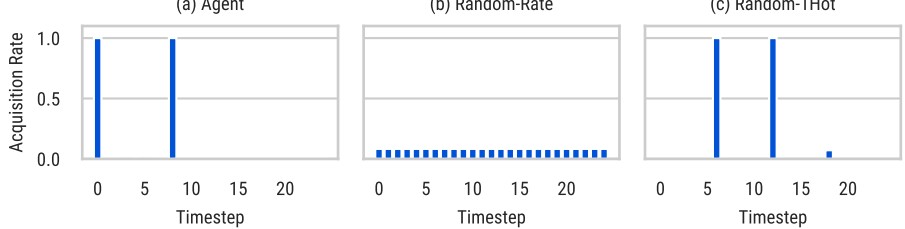

Figure 5: Comparing learned acquisition patterns of an agent on AudioSet to the patterns of the random ablations. Our agent learns a set of fixed timesteps for which it always acquires, similar to the random-1hot baseline. In (a) and (c), acquisition rates are close to zero and too small to be visible for some timesteps.

the number of agent acquisitions is not an integer, and so we remove some acquisition probability for the last of the fixed timesteps.) See fig. 5 for an illustration of the ablations.

These are ablations rather than baselines, as they use the per-modality acquisition rates found by the agent, and thus incur the same cost as the agent. However, potentially unlike the agent, the ablations do not act adaptively: they acquire with the same fixed probabilities for each sequence. If we find that our agent can consistently outperform both ablations, this is supporting evidence for adaptive behavior in the agent, adjusting its acquisitions to the information in each input.

For AudioSet, the agent does not consistently outperform either ablation at low acquisition costs. However, the agent does tend to outperform the random-rate, but not the random-1hot, ablation at medium-to-high acquisition costs. Figure 5 displays the average *temporal acquisition pattern* of the agent and ablations for the audio modality of AudioSet at high acquisition costs. The agent learns a *discrete* pattern of acquisitions across timesteps, similar to the random-1hot baseline. This is advantageous in comparison to the fixed low rates of the random-rate baseline: due to the small value of the acquisition probabilities, the resulting Binomial distribution over acquisitions has significant mass at 0 acquisitions. Therefore, the random-rate baseline sometimes does not acquire anything at all, which has a large negative effect on its average performance.

It is encouraging that the agents learn discrete acquisition behavior, avoiding the drawbacks of fixed small acquisition rates. However, this also shows that the agents do not learn *individualized* predictions, and instead acquire at fixed timesteps, ignoring individual inputs for decision making. There is almost no variance in acquisition behavior for different test samples. For scenarios with higher learned acquisition rates—where the variance of fixed-rate acquisition is less disadvantageous—we find some agents do not learn 1-hot style acquisition patterns and instead behave more similarly to random-rate.

For Kinetics, we observe that our agents are able to outperform the random-rate *and* random-1hot ablations. However, instead of individualized behavior, figs. 6 and A.3 show that the agents learn a static pattern where they never acquire both modalities simultaneously. While this is interesting behavior that optimizes the objective better than the random ablations (presumably because at least one modality is always acquired), it falls short of our goal of *adaptive* acquisition.

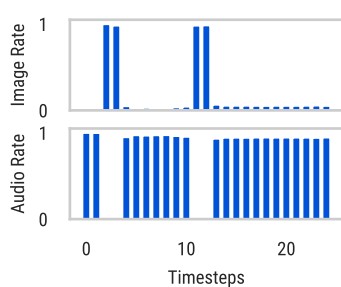

Figure 6: Acquisition patterns on Kinetics at 0.001 cost per image.

**Intermediate Rewards.** We investigate the effect of intermediate rewards on agent behavior. Figure 7 (a) displays 100 random samples of agent behavior from the AudioSet test set when using intermediate rewards. Intermediate rewards lead to agents that show *variable* acquisition behavior between samples. It also seems that intermediate reward—which encourages the agent to reduce predictive loss with each action—leads to greedy behavior in the agents, where acquisitions at earlier timesteps are preferred. Further, fig. 7 (b) shows the number of acquisitions per sample is now correlated to the entropy, which expresses certainty or uncertainty of the predictions. However, fig. 7 (c) shows that model entropy and loss are practically uncorrelated here. (We expand on this in our discussion in §7.) Comparing to the random ablations, we find that, while our AudioSet agents do learn adaptive behavior, their performance can still be matched by the non-adaptive ablations, cf. table A.5. For Kinetics, we observe similar behavior (except that we outperform the random ablations with the same caveat as above), cf. table A.6.

## 6 Related Work

**Additional Applications of A2MT.** A2MT-like problems can be found in a variety of application domains. For example in wearable devices, the energy cost of sensor activation is significant (Possas et al., 2018) and a policy that reduces sensor usage without sacrificing predictive accuracy is desirable. Literature in AFA further mentions computer security or fraud detection as areas of application. Given that both these areas naturally may have temporal or multimodal components, these applications transfer to A2MT as well. Lastly, the acquisition procedure in A2MT can be a way to reduce input size and thus computational cost of predictions.

**Active Perception.** Active perception (Bajcsy, 1988; Aloimonos et al., 1988; Bajcsy et al., 2018) models the cost associated with visual attention in the context of embodied agents. In the language of A2MT: if we cannot observe all visual input features at once, we need to decide where to look actively and iteratively. For

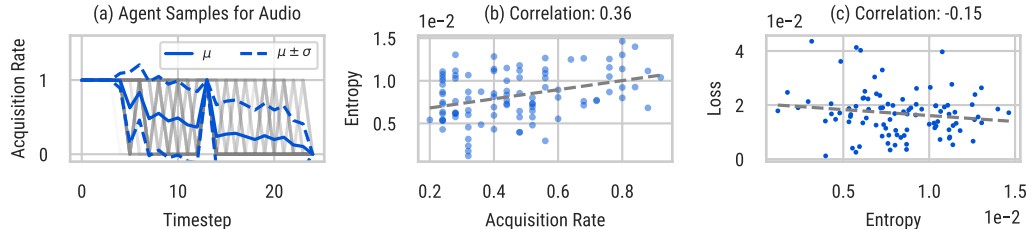

Figure 7: Behavior for agents trained with intermediate reward on 100 random samples from the AudioSet test set. (a) The acquisition behavior displays significant variance, and agents prefer acquisition at earlier timesteps. (Samples in grey, mean $\mu$ and standard deviation $\sigma$ in blue.) (b) The per-sample acquisition rates are correlated to the entropy of model predictions. (c) Correlation between entropy and predictive loss is poor.

example, Mnih et al. (2014); Haque et al. (2016); Jayaraman & Grauman (2018) solve visual classification or reconstruction problems by learning agents that iteratively attend to parts of the input. Relatedly, active sensing, e.g. Satsangi et al. (2018); Hero & Cochran (2011), studies how to actively select a subset of sensors to observe to reduce uncertainty over latent variables. Satsangi et al. (2020) connect model uncertainty and prediction reward in active perception.

**Efficient Video Classification.** Related work in efficient video classification has sought to reduce the computational cost of video classification by selecting a small set of salient frames for each input (Yeung et al., 2016; Wu et al., 2019b; Korbar et al., 2019; Wu et al., 2019a; Gao et al., 2020; Zheng et al., 2020; Wang et al., 2021; Ghodrati et al., 2021; Panda et al., 2021; Gowda et al., 2021; Yang et al., 2022). For computationally cheap policies, these methods can achieve computational cost-savings compared to methods that rely on all timesteps as input. This is related to so-called anytime prediction approaches (Grubb & Bagnell, 2012; Zilberstein, 1996; Horvitz, 1987) which explicitly trade off computation cost and predictive accuracy. In contrast to all of the above, A2MT assumes that there is a real-world cost associated with the *acquisition* of the input modalities, e.g. the cost of performing an MRI scan, and we can completely neglect any computational costs. Also, A2MT requires temporally causal acquisition, which is something not respected by most efficient video classification approaches. These difference in motivation lead to methods which (in all cases we know of) are not applicable to the A2MT scenario.

## 7 Discussion

While our agents did react sensibly to acquisition cost on the complex audio-visual datasets in §5.2.2, they learned static behavior and did not adjust acquisitions sensibly to individual sequence. In this section, we offer a discussion of possible reasons for this negative result: Figure 7 (c) suggest that our models may not be suitable for learning adaptive behavior. We would expect the uncertainty (entropy) of the predictions to correlate with the predictive loss. Then, entropy could be a useful signal to guide agent behavior: for samples where the model is certain about the prediction (low entropy) the agent can stop acquisitions early; when the model is uncertain, the agent may acquire more to reduce uncertainty and therefore loss. We do not see this correlation in fig. 7 (c), and so it may be hard for the agent to learn a policy that optimizes the reward, which depends on the pre-trained predictive model.

One possible explanation for the lack of correlation is that our Perceiver IO models currently underfit the AudioSet and Kinetics dataset. However, as we are not aware of prior work studying the entropies of Perceiver IO predictions, we cannot exclude the possibility that the entropies of Perceiver IO generally do not conform to expectations. Therefore, both training the initial model longer and additional model architectures may be worth exploring. Lastly, future work should not exclude the possibility that a subtle distribution shift in masking distributions between pre-training and agent training further inhibits policy learning.

Alternatively, AudioSet and Kinetics may be ill-suited for application to A2MT: while we have found that there is some signal diversity in AudioSet and Kinetics, different sections of a given clip often look similar, presumably making it difficult for the agents to learn adaptive behavior. Subsequent work in A2MT could consider datasets with longer clip durations and more content diversity per sequence, e.g. ActivityNet (Fabian Caba Heilbron & Niebles, 2015). Lastly, there are interesting variations of the A2MT setup that future work could explore, e.g. other reward formulations such as a fixed budget per sample or a global budget across samples, acquisition costs that change with time, or actions that affect state evolution.

## 8 Conclusion

We have introduced the task of active acquisition for multimodal temporal data (A2MT), extending prior work in active feature acquisition to multimodal and temporal inputs. We have further proposed a Perceiver IO–based reinforcement learning approach to tackle A2MT problems. On novel synthetic scenarios our agents successfully learn to use cross-modal reasoning to optimize the trade-off between feature acquisition cost and predictive accuracy. We further adapt Kinetics and AudioSet, two large scale video classification datasets, to application to A2MT: here, the agents appropriately react to modality-specific acquisition costs. However, ablations reveal they do not adapt acquisitions to individual inputs. We believe A2MT is a challenging and practically relevant task, and we would be excited for the community to join our efforts.

**Broader Impact Statement**

Our paper proposes a sequential decision making task, as well as a method to approach this task. We believe that this method can have useful societal and economic impact in domains such as robotics, finance, and healthcare. However, one of the main limitations of this paper is that we have focused on synthetic data scenarios. Although this synthetic data is motivated by real-world scenarios, we do not recommend direct deployment of our method to practical scenarios. Work on automated decision making should always be carried out in close collaboration with domain experts, while proactively taking into account safety and ethical considerations.

**Acknowledgments**

We thank Yujia Li, Timothy Lillicrap, Andrew Brock, Adrià Recasens, João Carreira, Lucas Smaira, Isabela Albuquerque, Joost van Amersfoort, Ali Eslami, our action editor, Martha White, and the anonymous reviewers for helpful feedback and interesting discussions that have led to numerous improvements of the paper.

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

# A   Additional Results

In fig. A.1, we display the training dynamics for the experiments on the synthetic scenario with raw inputs, fig. A.2 gives performance for the MNIST and the MNIST + Spoken Digit version of the synthetic experiment.

In tables A.1 and A.2, we report confusion matrices comparing our agent's acquisition strategy to oracle behavior on the raw version of the synthetic data experiment. Our test set has $10^4$ sequences of length $T = 10$, so there are a total of $10^5$ timesteps per modality for which we evaluate agent acquisitions. For the `digit` modality, optimal behavior is to acquire only when the `counter` modality is zero. Table A.1 shows the agent largely follows this optimal behavior, acquiring in 98.6% of cases where the counter is zero (true positives), and when the counter is not zero, the agent does not acquire in 84.8% of cases (true negatives). For the `counter` modality, oracle behavior is to acquire the first element of each countdown, as all values following as well as the start of the next countdown can be inferred. Because countdowns have a minimum length of 2, the `counter` does not need to be acquired when a new countdown starts at the last timestep. Table A.2 shows that agent behavior is less ideal for the `counter` modality, acquiring 71.6% of starting values (true positives) but not acquiring only for 33.0% of negatives (true negatives). In other words, at timesteps where the agent needs not acquire (negatives), the agent unnecessarily acquires 67.0% of the time (false positives). These extra acquisitions may help the agent better discover zeros in the counter modality when it misses a countdown starting value.

Table A.3 and table A.4 give results of our agents on AudioSet and Kinetics respectively when costs are imposed on the weaker input modality. For AudioSet, the agents learn to acquire $\approx 1/25$ image per sequence across a magnitude of acquisition costs. Surprisingly, the performance at 1 image frame is only about 1 p.p. mAP lower than the performance on fully observed data, which explains why it makes sense for the agent to quickly drop to such a low acquisition rate on the image modality. At high costs of 0.01 per acquisition, the agent no longer acquires any images, which finally does hurt mAP by about 4 p.p. For Kinetics, we observe that the number of acquisitions for the weaker audio modality drops steadily as cost is increased, and, correspondingly, so does the top1-accuracy. This indicates that, for Kinetics, the model makes better use of the weaker audio modality.

Figure A.3 shows the agent never acquires both modalities on the Kinetics dataset across a variety of costs.

Table A.5 and table A.6 give agent behavior with intermediate rewards enabled for AudioSet and Kinetics. They do not lead to significantly improved agent performance relative to the non-adaptive ablations.

# B   Synthetic Sequence Generation

Algorithm B.1 gives Python code for generating the sequences for the synthetic dataset (cf. §3.1).

---

**Algorithm B.1** Generate Synthetic Sequence

---

```python
def draw_sequence(length, digit_low, digit_high, counter_low, counter_high):

  digits, counter, important_digits = [], [], []

  while len(digits) < length:
    max_count = np.random.randint(counter_low + 1, counter_high + 1)
    counter_i = np.arange(max_count, counter_low - 1, -1)
    counter.extend(counter_i)

    digits_i = np.random.randint(digit_low, digit_high + 1, len(counter_i))
    if len(digits_i) + len(digits) <= length:
      important_digits.append(digits_i[-1])
    digits.extend(digits_i)

  digits = digits[:length]
  counter = counter[:length]
  label = sum(important_digits)

  return digits, counter, label
```

---

Table A.1: Confusion matrix for agent acquisitions of the `digit` modality compared to oracle behavior on the test set of the raw synthetic scenario.

|  |  | Oracle | |
|---|---|---|---|
|  |  | False | True |
| **Agent** | False | 53247 | 9575 |
|  | True | 528 | 36650 |

Table A.2: Confusion matrix for agent acquisitions of the `counter` modality compared to oracle behavior on the test set of the raw synthetic scenario.

|  |  | Oracle | |
|---|---|---|---|
|  |  | False | True |
| **Agent** | False | 19987 | 40665 |
|  | True | 11171 | 28177 |

Figure A.1: Performance of the Perceiver IO–based model and agent when applied to the raw-digit version of the synthetic dataset. The model learns to quickly solve the task, and the agent slowly reduces the number of acquired datapoints. Note that the 'Acquisitions' plots give acquisition rates averaged across time, s.t. a '1' corresponds to the entire modality being acquired.

## C  Experiment Details

### C.1  Synthetic Experiments: Raw Inputs

#### C.1.1  Dataset

We generate a synthetic dataset with training set of size $50 \times 10^3$ and test set of size $10 \times 10^3$. For the 'raw' version, the input modalities have shape $(10, 3)$, i.e. we use sequences of length 10 and there are three distinct values $(2, 1, 0)$ per modality. We set the acquisition cost to 0.0005 per modality and timestep.

#### C.1.2  Architecture

We train using a batch size of 256. We use the ADAM optimizer with initial learning rate of $3 \times 10^{-4}$, weight decay of $1 \times 10^{-6}$, and a cosine annealing schedule. For the Perceiver IO encoder we use a single cross-attend block with 4 self-attention operations per Perceiver IO block; we use 128 queries, and the hidden dimension is 128. For the Perceiver IO decoder, we use a single head with 128 queries and hidden dim of 128. We train for a total of $2 \times 10^5$ steps. We set the discount factor to $\gamma = 1$.

#### C.1.3  Choosing Acquisition Costs

In this section, we detail how we arrived at our selection of acquisition costs for the results in §5.2.2. For both AudioSet and Kinetics, we found valid cost ranges through experimentation. As a rule of thumb, we aimed for acquisition costs such that $T$ times the cost is smaller than the observed predictive loss on the fully unmasked data. If this were not the case, it is too attractive to not acquire any input samples. Note that predictive losses between AudioSet and Kinetics are significantly different; AudioSet is a multi-class (per instance) problem and Kinetics is not. As costs are directly traded off with predictive loss in our objective, cf. eq. (1), this explains the different cost magnitudes between the datasets.

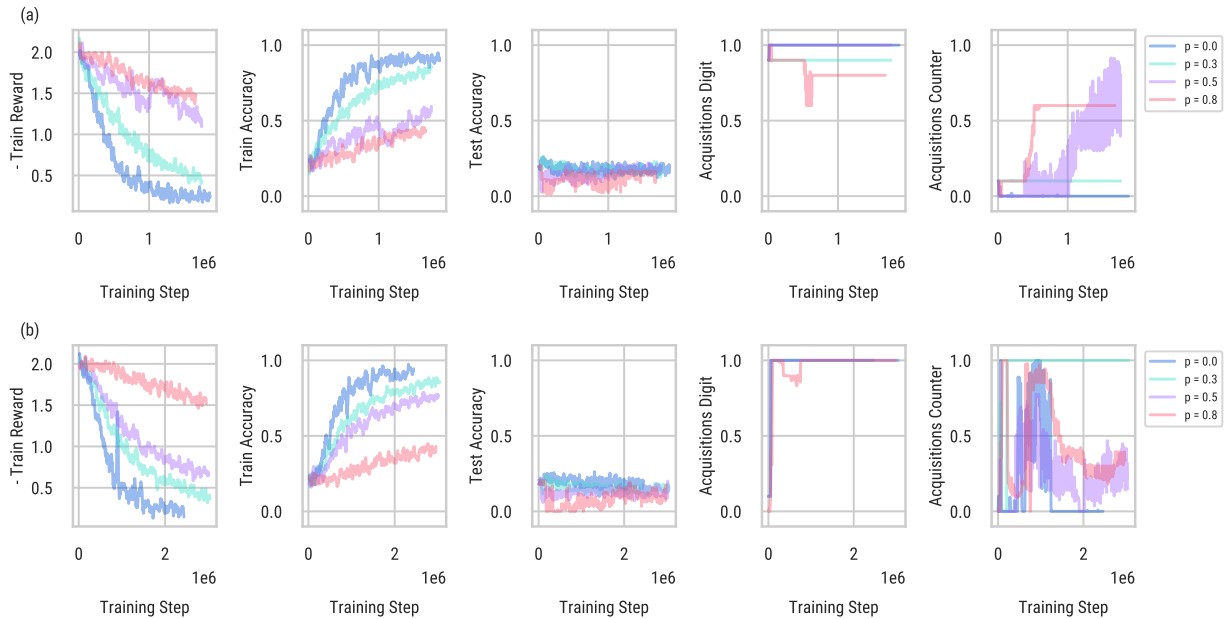

Figure A.2: Performance of the Perceiver IO–based model when applied to the (a) MNIST and (b) MNIST/Spoken Digit version of the synthetic dataset. The agents are unable to learn prediction strategies that generalize to the test set for this scenario. The different plots show a sweep over dropout probabilities in $[0, 0.3, 0.5, 0.8]$ for the input tokens: while we do observe a regularizing effect for dropout here, it does not lead to improved generalization performance. Acquisition cost is set to 0 for these runs. Note that the 'Acquisitions' plots give acquisition rates s.t. a '1' corresponds to the entire modality being acquired. We apply running mean smoothing with kernel size 10 for the plots of the first two columns.

Table A.3: Acquisition behavior for the agent and ablations as costs increase for the image modality of the AudioSet dataset. Standard deviations over 5 repetitions over the test set.

| Cost Image | 0.00001 | 0.00010 | 0.00100 |
|---|---|---|---|
| Agent Acquisition Rate | 0.07 | 0.05 | 0.01 |
| Agent (mAP) | $0.3683 \pm 0.0060$ | $0.3627 \pm 0.0017$ | $0.3289 \pm 0.0072$ |
| Random-Rate (mAP) | $0.3505 \pm 0.0005$ | $0.3470 \pm 0.0010$ | $0.3238 \pm 0.0008$ |
| Random-1Hot (mAP) | $0.3681 \pm 0.0003$ | $0.3650 \pm 0.0004$ | $0.3275 \pm 0.0004$ |

Table A.4: Acquisition behavior for the agent and ablations as costs increase for the audio modality of the Kinetics dataset. Standard deviations over 5 repetitions over the test set.

| Cost Audio | 0.001 | 0.010 | 0.100 |
|---|---|---|---|
| Agent Acquisition Rate | 0.79 | 0.43 | 0.05 |
| Agent (top1) | $0.3313 \pm 0.0041$ | $0.3253 \pm 0.0036$ | $0.2672 \pm 0.0015$ |
| Random-Rate (top1) | $0.2825 \pm 0.0012$ | $0.2840 \pm 0.0020$ | $0.2524 \pm 0.0013$ |
| Random-1Hot (top1) | $0.3063 \pm 0.0003$ | $0.2930 \pm 0.0003$ | $0.2638 \pm 0.0004$ |

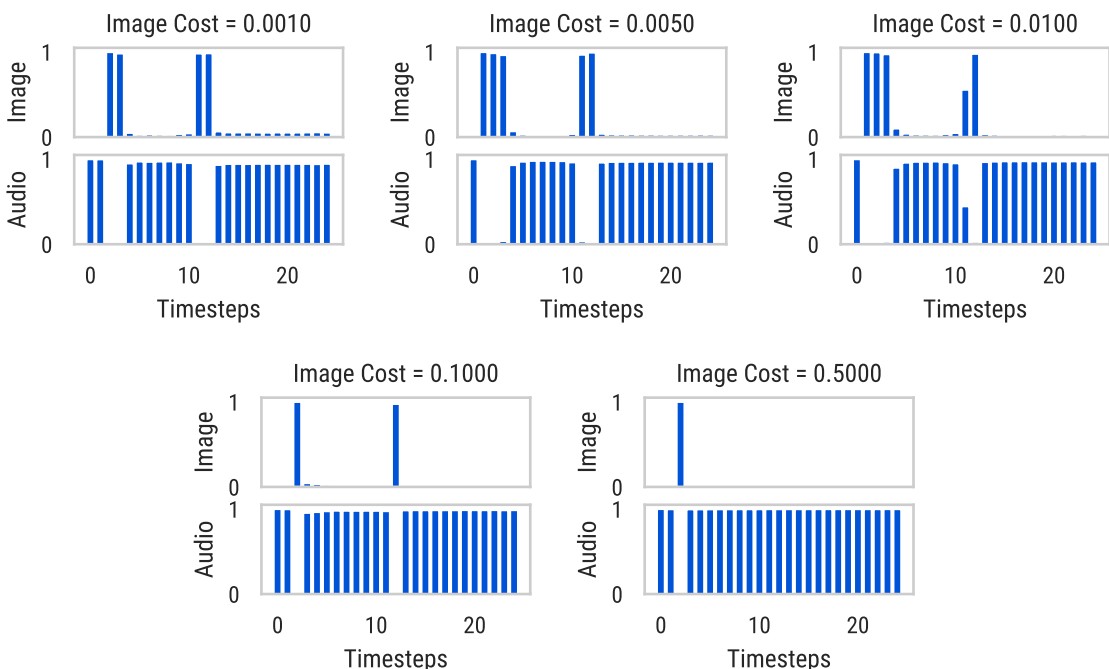

Figure A.3: Learned patterns on Kinetics at a variety of image cost. The y-axis gives the average acquisition rate of the agent for a modality and timestep. As the image cost increases, the number of acquisitions decreases. Further, the agents learn a static pattern where at least one modality is always acquired.

Table A.5: Agent behavior with intermediate rewards on AudioSet with different audio acquisition costs and trade-off parameters $\alpha$. See §4.2 for an explanation of intermediate rewards and $\alpha$. Standard deviations over 5 repetitions over the test set.

| Cost Audio | 0.0005 | 0.0005 | 0.00010 | 0.00010 |
| Trade-off $\alpha$ | 0.1 | 0.5 | 0.1 | 0.5 |
| --- | --- | --- | --- | --- |
| Agent Acquisition Rate | 0.05 | 0.13 | 0.04 | 0.06 |
| Agent (mAP) | $0.2648 \pm 0.0063$ | $0.3104 \pm 0.0120$ | $0.2671 \pm 0.0009$ | $0.2668 \pm 0.0075$ |
| Random-Rate (mAP) | $0.2439 \pm 0.0012$ | $0.3043 \pm 0.0013$ | $0.2323 \pm 0.0014$ | $0.2570 \pm 0.0014$ |
| Random-1Hot (mAP) | $0.2750 \pm 0.0006$ | $0.3238 \pm 0.0004$ | $0.2664 \pm 0.0003$ | $0.2838 \pm 0.0008$ |

Table A.6: Agent behavior with intermediate rewards on Kinetics with different image acquisition costs and trade-off parameters $\alpha$. See §4.2 for an explanation of intermediate rewards and $\alpha$. Standard deviations over 5 repetitions over the test set.

| Cost Image | 0.01 | 0.01 | 0.10 | 0.10 |
| Trade-off $\alpha$ | 0.1 | 0.5 | 0.1 | 0.5 |
| --- | --- | --- | --- | --- |
| Agent Acquisition Rate | 0.19 | 0.27 | 0.15 | 0.18 |
| Agent (top1) | $0.3260 \pm 0.0018$ | $0.3234 \pm 0.0012$ | $0.3225 \pm 0.0027$ | $0.3162 \pm 0.0008$ |
| Random-Rate (top1) | $0.2895 \pm 0.0004$ | $0.2673 \pm 0.0011$ | $0.2768 \pm 0.0011$ | $0.2807 \pm 0.0009$ |
| Random-1Hot (top1) | $0.3015 \pm 0.0003$ | $0.2798 \pm 0.0004$ | $0.3042 \pm 0.0004$ | $0.3037 \pm 0.0003$ |

Once we found a valid cost value, we increased and decreased costs to find a range of costs that leads to varied agent behavior. First, we increased costs until the agent (almost) did not make any acquisitions anymore. Tables 5 and 6 show this requires adjusting the costs across multiple magnitudes. Note that acquisition rates of about 0.04 correspond to acquiring a single segment for our total of $T = 25$ segments. As the agent already acquires only a single segment, additional cost increases are therefore not that interesting from the perspective of A2MT. We also decreased costs until agent behavior became stagnant. For AudioSet at the lowest cost, the agent acquires (almost) all segments of the audio signal, so further decreasing the cost would not change behavior. For Kinetics, agent acquisition rates are already constant for the three lowest acquisition costs, and it is unlikely that further cost decreases would change this. It seems that the agent refuses to learn to acquire more than 20% of the image modality, regardless of how low the cost is. In table 4, we observed that predictive performance on Kinetics is almost unchanged until mask rates are increased above 80%, which points towards the repetitive nature of the image modality in the Kinetics dataset. It is therefore likely that the agent behavior is sensible here, as there is simply no benefit to increasing acquisitions above 20% for the image modality of Kinetics for the Perceiver IO predictive model.

## C.2 Synthetic Experiments: MNIST and SpokenDigit Versions

For all hyperparameters not mentioned in the below, we use the same settings as for the 'raw' version of the synthetic dataset. We use a batch size of 128 and train for $> 1 \times 10^6$ steps. The input shape of the MNIST images is $(10, 28, 28)$, and the shape of the audio snippets is $(10, 39, 80)$ after mel spectrogram pre-processing.

## C.3 Audio-Visual Datasets

### C.3.1 Dataset

The AudioSet dataset has a training set of size $1\,771\,873$, an evaluation set of size $17\,748$, and 632 classes. We use the unbalanced version of the dataset. The Kinetics dataset has a training set of size $545\,793$, a test set of size $67\,858$, and 700 classes. For both datasets, each input sample consists of audio-visual input with 250 frames of images and a raw audio signal spanning 10 seconds. For images, we take every 10th frame as input, obtaining a total of 25 input frames. We pre-process the audio signal to mel spectrograms, and then divide the signal into 25 segments. The audio modality has input shape $(25, 40, 128)$, and the image modality has input shape $(25, 200, 200, 3)$.

## C.4 Architecture

**Perceiver IO.** For the Perceiver IO encoder we use a single cross-attend block with 8 self-attention operations per Perceiver IO block; we use 1024 queries, and the hidden dimension is 1024. For the Perceiver IO decoder, i.e. the policy head, we use a single cross-attend head with 1024 queries and hidden dim of 1024. The A2C baseline network is also a Perceiver IO decoder module with the same configuration as the policy.

**Pre-Training.** We use the ADAM optimizer with initial learning rate of $3 \times 10^{-4}$, weight decay of $1 \times 10^{-6}$, and a cosine annealing schedule. We train for a total of 100 epochs using a batch size of 512. We use the masking settings detailed below.

For the masking variants **M1-M3**, we report results with the following settings. For **M1**, we keep inputs at given modality and timestep with $p_m = 0.2$. For **M2**, we set $p_m = 0.4$, which accounting for the additional 'max-timestep' masking mechanism (which masks out half the input on average), yields the same expected unmasking rate of 0.2. For **M3**, we keep $p_m = 0.4$ and additionally drop modalities with $d_m = 0.5$. In all of the above, probabilities are the same across modalities $m$. These settings performed best in preliminary experiments.

**Agent Training.** We train agents for a total of $20 \times 10^3$ steps. We re-use the optimizer configuration from pre-training for training of the agent and the A2C baseline network. We set the discount factor to $\gamma = 1$.

# D   A2MT as a Markov Decision Process

Here, we attempt to formally connect A2MT to Markov Decision Processes (MDPs) which are the main paradigm for framing problems in the reinforcement learning literature (Sutton & Barto, 2018). Concretely, we believe that a Partially Observable MDP (POMDP) is best suited to the A2MT scenario. We refer to Yin et al. (2020); Janisch et al. (2019; 2020) for a similar treatment of the (PO)MDP action space for active feature acquisition problems.

POMDPs are defined by the 6-tuple $(\mathcal{S}, \mathcal{A}, \mathcal{P}, \mathcal{R}, \Omega, \mathcal{O})$. At each timestep $t$, the environment is in some state $\boldsymbol{s}_t \in \mathcal{S}$ and the agent selects an action $\boldsymbol{a}_t \in \mathcal{A}$. This leads to a new state $\boldsymbol{s}_{t+1} \sim \mathcal{P}(\cdot|\boldsymbol{s}_t, \boldsymbol{a}_t)$ and reward $r_t \sim \mathcal{R}(\boldsymbol{s}_t, \boldsymbol{a}_t)$. In a POMDP, the agent never observes states directly, and instead has access only to observations $\boldsymbol{o}_t \in \Omega$ depending on the unobserved state, $\boldsymbol{o}_t \sim \mathcal{O}(\cdot|\boldsymbol{s}_t)$. To select actions, the agent uses a policy depending only on the current observation $\boldsymbol{a}_t \sim \pi(\cdot|\boldsymbol{o}_t)$.

Using the notation introduced above and in §2, we can align the A2MT framework with a POMDP by making the following identifications: (1) Actions for $t \in (1, \ldots, T)$ are binary acquisition decisions per modality, $\boldsymbol{a}_t \in \mathcal{A}^B, t \in (1, \ldots, T)$. After the sequence is consumed, at $t = T + 1$, we insert an additional timestep at which no action is taken, $\mathcal{A}^P = \emptyset$. Formally, the action space $\mathcal{A}$ is the union of both spaces, $\mathcal{A} = \mathcal{A}^B \cup \mathcal{A}^P$. (2) For $t \in (1, \ldots, T)$, the state $\boldsymbol{s}_t$ contains the unmasked sequence of the modalities until time $t$, i.e. the *sequence* of observations $\boldsymbol{x}_{1:t}$, as well as a vector of all previous agent actions, $\boldsymbol{a}_{1:t}$. Here, the modalities $p(\boldsymbol{x}_t|\boldsymbol{x}_{t+1})$ evolve according to the data generating distribution, e.g. the algorithm for the synthetic dataset creation in §3.1 or the generative model underlying AudioSet/Kinetics video data. At time $T + 1$, the state additionally contains the label associated with the multimodal sequence, i.e. a sample from the conditional distribution $p(y|\boldsymbol{x}_{1:T})$ of the generating process. (3) The observation kernel $\mathcal{O}(\boldsymbol{o}_t|\boldsymbol{s}_t)$ emits a sequence of (partially) masked modalities according to the acquisition pattern of the agent up to time $t$, $\boldsymbol{o}_t = \tilde{\boldsymbol{x}}_{1:t}$. Note that because of our construction in (2), the state contains all necessary information to assemble this sequence at each timestep. No observation is emitted at $o_{T+1}$. (4) For $t \in (1, \ldots, T)$, the reward function is defined as $R_t(\boldsymbol{s}_t, \boldsymbol{a}_t) = R_t(\boldsymbol{a}_t) = -\sum_m c_m a_{t,m}$, i.e. the reward is the negative modality-specific acquisition cost at that timestep, which does not depend on the state, cf. eq. (1). For $t = T + 1$, the reward is given by the negative loss achieved by the classifier $R(\boldsymbol{s}_{T+1}, \boldsymbol{a}_{T+1}) = R(\boldsymbol{s}_{T+1}) = -\mathcal{L}(f(\tilde{\boldsymbol{x}}_{1:T}), y)$. Note that, instead of modelling the prediction $f(\tilde{\boldsymbol{x}}_{1:T}), y)$ as an action, we include $f$ as part of the global environment state as it is not trained by reinforcement learning. When $f$ is fixed, e.g. in our large scale experiments, the reward from the classification at $T + 1$ depends only on the acquisitions of the agent; $f$ and its parameters are fixed across sequences. When $f$ is trained jointly with the policy, the parameters of $f$ are updated between—but not within—episodes, making the reward and thus the POMDP nonstationary.

