# OpenReview forum: "Active Acquisition for Multimodal Temporal Data: A Challenging Decision-Making Task"
_TMLR — Accepted by TMLR_

### Review · Reviewer_sDJ9 · 2023-04-29

**Summary Of Contributions:**

- The paper introduces a novel task formulation, i.e., the Active Acquisition for Multimodal Temporal Data (A2MT) problem.
- The authors have designed synthetic scenarios and validated on real-world datasets, AudioSet and Kinetics.
- The paper introduces a reinforcement learning approach based on Perceiver IO to tackle the proposed problem with a thorough empirical
study. The agents' appropriate response to modality-specific acquisition costs demonstrates the potential of the proposed method for real-world applications.

**Audience:**

Yes

**Broader Impact Concerns:**

Not applicable.

**Claims And Evidence:**

Yes

**Requested Changes:**

Major:
- (See weakness for details) I would really like to see how the presented method works on a more continuous time-dependent synthetic scenario, e.g., a simple Markovian counter sequence.

Minor/Formatting:
- In Section 2, the notation $c$ has been used to denote both color channels and cost. Please consider differentiating them.
- Figure 5 and 7 are too small to read. Please consider enlarging them.

**Strengths And Weaknesses:**

Strengths:
- The motivation of the task formulation is clear and evident, and one can clearly see could have potential applications in various fields like finance, and healthcare.
- The formulated problem is novel and in time, which is an up-to-date formulation for the classic Active Feature Acquisition (AFA) problem.
- The empirical findings, especially those in Section 5.2.2, are of interest to the TMLR community.
- Overall, the delivery and presentation of the work are mostly clear and easy to follow, with some minor formatting issues that need to be addressed.

Weaknesses:
- The discussion of related work can improve, in particular, the authors should consider situating or differentiating their work against the line of work in multimodal active learning.
- Based on my understanding, the synthetic scenario used in the paper is not a natural abstraction of the temporal decision-making problem. Especially, I am not sure if the feature values evolve.
> "Notably, prior work in AFA assumes static data: although acquisitions are sequential, feature values do not evolve along a temporal dimension."
>
More specifically, the synthetic scenario involves a series of random counting sequences, in which the starting number does not depend on the previous sequences.
> "The counter modality repeatedly counts down from a randomly drawn starting value, e.g. 2103210210. The label attached to each sequence is the sum of the digit modality at all timesteps where the counter modality is zero, e.g. 3+3+3=9 in this case."
>
The presented problem is more of a reasoning problem over sequences than a temporal decision-making problem, to my understanding. For a more natural abstraction, a Markovian (or more complicated temporally dependent sequence) sequence would make more sense. I would love to hear more about the design choice of this experiment from the authors.

---

> ### Author Response · Authors · 2023-05-05
> **Author Response to Reviewer sDJ9 (Part 1)**
>
> Dear reviewer sDJ9,
>
> Thank you for your hard work and helpful feedback. We have gladly added a discussion of the relationship between A2MT and active learning and hope that you will engage with us actively during the discussion period to clarify any remaining points, particularly about the synthetic scenario. Please read our comment above, addressed to all reviewers, first.
>
>
> > The discussion of related work can improve, in particular, the authors should consider situating their work against the line of work in multimodal active learning.
>
>
> We are happy to discuss the relation of A2MT and active learning in our paper. We have added a sentence in the introduction to clarify the fundamental differences between active learning (AL) and active feature acquisition / A2MT.
>
>
> AL minimizes the number of _label_ acquisitions needed for model _training_. A2MT minimizes the number of _feature acquisitions_ for _a given sample_ to make a prediction. So while AL concerns itself with label acquisition for model training, A2MT (and AFA) usually assumes that the data are fully observed during training. And while AL iteratively decides for which training point to acquire a label, A2MT makes iterative acquisition decisions about the features given a single test point.
>
>
> These fundamental differences hold true for multimodal data as well.
>
>
> In terms of literature on multimodal active learning, would you happen to have any suggestions?
> We have done some quick research and could only find [A] and [B].  Despite their matching title, neither really fit well.
>
>
> [A] focuses on a specific application “real-world child-robot interactions during an autism therapy”. Further [A] applies heavy preprocessing to the input images and sound (openFace, openPose, openSMILE) such that it may perhaps be misleading to call the resulting low-dimensional inputs multi-modal; just like it would be misleading to call active learning on tabular data ‘multimodal’ because different columns represent different modalities.
>
>
> [B] focuses on “Multimedia Streaming Applications”. Video streaming is not our domain of expertise, however, at first glance, the setup in [B] seems so far removed from standard active learning, that it might be misleading to cite it as such.
>
>
> Perhaps the lack of literature can be explained by the fact that active learning is much more straightforward to apply to the multimodal setting than AFA/A2MT.
> For active learning, we typically only need a predictive model $y|x_a, x_b$ _conditioned_ on the multimodal features $x_a, x_b$, whereas AFA/A2MT need to reason about the multimodal features themselves, i.e. $x_a|x_b$. In other words, for active learning on multimodal data, one can continue to use the same tools from standard active learning.
> Please let us know if you think we missed any important literature on multimodal active learning.
>
>
>
> [A]: Rudovic, O., Zhang, M., Schuller, B., & Picard, R. (2019, October). Multi-modal active learning from human data: A deep reinforcement learning approach. In 2019 International Conference on Multimodal Interaction (pp. 6-15).
>
>
> [B] Dhiman, G., Kumar, A.V., Nirmalan, R. et al. Multi-modal active learning with deep reinforcement learning for target feature extraction in multi-media image processing applications. Multimed Tools Appl 82, 5343–5367 (2023).
>
>
> > Based on my understanding, the synthetic scenario used in the paper is not a natural abstraction of the temporal decision-making problem.
>
>
> Thanks for raising this! First, we would like to make sure that there is no misunderstanding here about the nature of the synthetic scenario. In particular, we would like to highlight the construction of the `counter` modality. We have also clarified this in the revised draft!
>
>
> We draw multiple starting values and then concatenate the countdown sequences, i.e. the starting values `[2,3,2]` lead to the countdowns `[[2,1,0],[3,2,1,0],[2,1,0]` which are then concatenated to yield the `counter` modality `[2,1,0,3,2,1,0,2,1,0]`.  The counter modality is randomly generated and different between each input sequence.
>
>
> If there was a misunderstanding of the construction of the counter modality, we hope that our explanation above was able to alleviate any concerns, and that our clarifications allow you to set the 'Claims and Evidence' category to 'Yes'.
>
>
> If there was no misunderstanding, we would like respectfully disagree with some of the statements you make in your review.
>
>
> **Please read part 2 of our reply next.**

---

> > ### Author Response · Authors · 2023-05-05
> > **Author Response to Reviewer sDJ9 (Part 2)**
> >
> > **Please read part 1 of our reply first.**
> >
> > > Especially, I am not sure if the feature values evolve. [...]
> >
> >
> > Could you clarify what exactly you mean here? While the individual starting values are independent, the ‘countdowns’, and therefore the `counter` modality as a whole, has clear temporal dependence. The features of the counter modality ‘evolve’ across time. In fact, the counter modality at $t-1$ is fully predictive of the counter modality at $t$, unless the counter is $0$ at $t-1$ (which would trigger the next countdown).
> >
> >
> > > Especially, I am not sure if the feature values evolve. [...]  More specifically, the synthetic scenario involves a series of random counting sequences, in which the **starting number** does not depend on the **previous sequences** [highlights ours].
> >
> >
> > Reading these two sentences together, it might be possible there is another misunderstanding here.
> >
> >
> > First, we want to highlight that we draw _multiple_ starting numbers per sequence, and that starting numbers are randomly drawn for each sequence.
> >
> >
> > Secondly, by **previous sequences**, did you mean previous timesteps within a sequence, or previous sequences as in ‘entirely different episodes’ for the agent?
> > If you are referring to the latter, this is entirely by design!
> > A2MT assumes that different sequences/episodes are _independent_ and that feature values do not evolve between different sequences, only temporally within a given sequence.
> > Feature distributions that change between sequences are out of scope for A2MT (and AFA more generally).
> >
> >
> > We would be happy to make this clarification in our definition of A2MT if you think this could be a point of confusion.
> >
> >
> > > [...] The presented problem is more of a reasoning problem over sequences than a temporal decision-making problem, to my understanding.
> >
> >
> > We are not sure how you distinguish ‘reasoning problem over sequences’ from ‘temporal decision-making problem’.
> > From our perspective, the A2MT agent does not get to ‘reason over the sequence’ as a whole. Instead has to make a decision at each timestep (temporal decision making).
> >
> >
> > Further, this decision _will_ depend on previous observations in the sequence, in particular, if the agent learns to skip acquisitions of the counter modality to save on acquisition cost (which it does!).
> > For example, if the agent observes a counter value of $x=2$ at $t=0$, it does not have to acquire the counter modality at $t=1$ or $t=2$, because it can infer the counter values from $x$ at $t=0$.
> > Doing this, requires reasoning about the *temporal evolution* of the counter modality per input sequence.
> > Note that we _do_ observe this behavior emerging (cf. Figure 4).
> >
> >
> > In summary, we would argue that the following are true in the synthetic scenario: (1) ‘feature values evolve’ temporally in a given sequence and (2) the synthetic scenario is a ‘temporal decision-making problem’.
> >
> >
> > Please let us know if this has cleared up a potential confusion or if you think we have misunderstood your comment in any of the above.
> >
> >
> > > For a more natural abstraction, a Markovian (or more complicated temporally dependent sequence) sequence would make more sense.
> > >  I would really like to see how the presented method works on a more continuous time-dependent synthetic scenario, e.g., a simple Markovian counter sequence.
> >
> >
> > If your criticism still stands after our reply above, we would be happy to brainstorm additional synthetic scenarios with you.
> > For now, we continue to believe that our synthetic scenario already constitutes a temporally dependent sequence.
> >
> >
> > Could you perhaps elaborate what you mean by ‘Markovian counter sequence’; did you just mean Markov chain? (Did you have any more specific thoughts on how to integrate a cost structure? One of benefits of our current synthetic scenario, is that it is really simple to visually compare agent behavior to optimal acquisition.)

---

> > > ### Comment · Reviewer_sDJ9 · 2023-05-11
> > > **Reviewer Response to the Authors**
> > >
> > > Dear authors,
> > >
> > > Thank you for your excellent work in addressing my concerns!
> > >
> > > First of all, I found the following clarification regarding the differences between AL and AFA particularly helpful, which helps readers to better understand how this problem formulation is different from a typical AL setup. It would be great if a brief discussion on this difference can be presented in the next iteration of this paper.
> > >
> > > > Perhaps the lack of literature can be explained by the fact that active learning is much more straightforward to apply to the multimodal setting than AFA/A2MT... from standard active learning.
> > >
> > > Indeed, I still believe that the synthetic scenario used in the paper is not a **natural** abstraction of the temporal decision-making problem. Specifically,
> > > > We draw multiple starting values and then concatenate the countdown sequences, i.e. the starting values `[2,3,2]` lead to the countdowns `[[2,1,0],[3,2,1,0],[2,1,0]` which are then concatenated to yield the counter modality `[2,1,0,3,2,1,0,2,1,0]`. The counter modality is randomly generated and different between each input sequence.
> > >
> > > If I understand it correctly, the starting values `[2,3,2]` are randomly determined, which does not depend on the previous starting values. (Please correct me if I am wrong!) My concern is that this scenario does not seem like a noisy but natural abstraction of a real multimodal decision-making problem. When I mention ‘Markovian counter sequence’, I indeed mean a counter sequence whose values are determined by a Markov process rather than random. To me, this is a better abstraction.
> > >
> > > However, I do agree that:
> > > > One of benefits of our current synthetic scenario, is that it is really simple to visually compare agent behavior to optimal acquisition.
> > >
> > > Although I still believe a more systematic design of additional synthetic scenarios would be valuable, I understand the current setup and think this is very convincing to me as it might be hard to design a thorough set of Markovian sequences to study the problem.
> > >
> > > Overall, most of my concerns have been clearly addressed by the authors.

---

> > > > ### Author Response · Authors · 2023-05-11
> > > > **Author Response to Reviewer**
> > > >
> > > > We are very happy to hear that our replies and revisions were able to address your concerns.
> > > >
> > > >
> > > > Thank you for clarifying your idea of the Markovian counter sequence. We agree that this could be an interesting direction for more natural synthetic scenarios. However, we agree they 'might be hard to design' (and evaluate) and are happy that you find the current synthetic scenario 'very convincing'!
> > > >
> > > >
> > > > We will gladly add a brief discussion of the differences between AL and AFA to the next iteration of the paper.
> > > >
> > > >
> > > > Lastly, we would like to thank you for engaging with us during the rebuttal period. We really believe these interactions greatly enhance the experience and quality of peer review.

---

### Review · Reviewer_5VUY · 2023-05-01

**Summary Of Contributions:**

This paper presents a framework for classification in which costs are assigned to observing different multimodal inputs (e.g. images or audio), and implements an approach to perform this classification with a Perceiver-IO type of architecture that takes actions that determine whether it attends to each of the different inputs. Experiments on a controlled synthetic dataset and several larger datasets evaluate the methods performance relative to its ablations. The paper finds that in some settings on some datasets, the method outperforms ablations in which the attention to the inputs is set at a fixed rate and at a fixed frequency.

**Audience:**

Yes

**Claims And Evidence:**

Yes

**Requested Changes:**

Critical weaknesses to address: W1.1, W2.2 (cost-insensitive prior work), W2.3.
The remaining weaknesses are important (including (W2.2 cost-sensitive prior work)) but noncritical.

**Strengths And Weaknesses:**

## Strengths
- The paper proposes an interesting task and datasets on which to evaluate it. The task generalizes some existing tasks that either assume that the acquirable features do not vary in time or that the acquirable features are low-dimensional.
- The paper evaluates relative to some very important ablations and presents findings despite their "nonsignificance" in many settings, which informs readers that there is significant room for improvement on the methods side.
- The paper evaluates on a tightly controlled synthetic regime, which is also itself a good regime for further experimentation on varying some of the fundamental characteristics of the data and evaluation.

## Weaknesses
### 1: Presentation weaknesses
- [W1.1] would be significantly improved with an architecture diagram of the policy and the classifier.
- [W1.2] S3.1: It is unclear if the randomly drawn starting value for the counter is drawn once or multiple times. The example sequence given: “2103210210” appears to be only realizable if it is drawn multiple times, assuming values: [2, 3, 2]. Please clarify this in the paper.
- [W1.3] The experiment presented in Table 3 is unclear. A mask rate of 0 results in AudioSet mAP of 0.370, which appears to correspond to the 0.370 values in Table 2 and Table 1. However, the 0.370 value in Table 1 includes masking (M1,M2,M3). It is unclear what a “Mask Rate” of 0.0 (Table 3 Row 1) and masking of M1, M2, and M3 combined are doing, because masking with 0.0 probability seems like a contradiction with applying the masking of M1, M2, and M3. Is the mask rate the input to M1, M2, and M3? This would make sense if it were used as the parameter of the Bernoulli distribution, so it would work for M1 and M3, but not M2. My main guess here is that the Mask Rate in Table 3 is the rate of masking the test data, not the rate of masking the training data. Please clarify the paper so that others are less likely to have to guess at this. If my guess is correct, I suggest using terminology that clarifies the difference between masking when it is used as model optimization hyperparameter, vs. when it is used as a (test/eval) dataset parameter.
- [W1.4] S5.1 would be improved by presenting the metrics in a table, rather than inline, in order to increase their visibility.
- [W1.5] The related work section could be improved by including discussion of the “anytime prediction” framework and approaches (e.g., but not limited to, [A,B,C,D] in the References section below), in which a learner’s output can be computed at “any time” and improves with the addition of more inputs. The proposed A2MT framework, in which a single prediction is made after a fixed number of timesteps with costs on using certain inputs, could be considered a degenerate version of anytime prediction, which also imposes costs on using certain inputs, but requires approaches to be capable of producing an output at every timestep.
### 2: Experimental weaknesses
- [W2.1] The experiments in S5.1 and Fig A.1 could be improved with more metrics of counter acquisition that are normalized relative to the optimal number of acquisitions and the precise timesteps an optimal agent would attend to the counter. The optimal number of acquisitions is determined by the counter variable(s) — the digits need only be attended to by an optimal agent where the counter is 0, so observing the counter values after each 0 is sufficient. The paper states that 68.6% of the counter modality is attended to, but it is unclear how much an optimal agent should be attending to the test data. Furthermore, matching the optimal attention rate to counter variables is insufficient for attending to the correct counter variables — the analysis could be further improved with a metric of accuracy of attention to the “necessary” counter variables — scoring true positives for attention to the counter values that succeed 0, false negatives for failing to attend to those values, and false positives for counter modality attention elsewhere. These same metrics could be applied to the digit modality. These types of accuracy metrics are particularly valuable for this synthetic dataset because they’re easily computable relative to exactly computing or estimating them for the Audio-Visual datasets.
- [W2.2] The experiments in Table 4 and Table 5 would be significantly improved if they included prior work on these same classification tasks, both from approaches that are cost-insensitive (e.g. any prior classification approaches, which I’m fairly confident exist) and those that are cost-sensitive, if they exist on these dataset (I don’t know such exist on these datasets). The former type of approaches would ground the proposed methods potential improvement over cost-insensitive approaches, which is perhaps the main value proposition of any A2MT-based method.
- [W2.3] It is unclear how the ranges of cost values investigated in Table 4 and Table 5 were chosen. The paper would be improved with additional experiments here. The cost ranges appear to differ significantly: in [1e-6, 1e-3] for audio, vs. [1e-3, 5e-1] for images. It is possible that the effect of outperforming the random ablations is only present in the lower range [1e-3, 5e-1] for both modalities, and that the effect of underperforming the ablations is only present in the higher range [1e-6, 1e-3]. If the experiments presented results that used the same dense ranges for both modalities, we could draw stronger conclusions.
### References
- [A] Zilberstein, Shlomo. "Using anytime algorithms in intelligent systems." AI magazine 17.3 (1996): 73-73.
- [B] Horvitz, Eric J. "Reasoning about beliefs and actions under computational resource constraints." arXiv preprint arXiv:1304.2759 (2013).
- [C] Shani, Guy, Joelle Pineau, and Robert Kaplow. "A survey of point-based POMDP solvers." Autonomous Agents and Multi-Agent Systems 27 (2013): 1-51.
- [D] Grubb, Alex, and Drew Bagnell. "Speedboost: Anytime prediction with uniform near-optimality." Artificial Intelligence and Statistics. PMLR, 2012.

---

> ### Author Response · Authors · 2023-05-05
> **Author Response to Reviewer 5VUY (Part 1)**
>
> Dear reviewer 5VUY,
>
>
> Thank you for your hard work and helpful feedback. We have very gladly incorporated many of your excellent suggestions and hope that you will engage with us actively during the discussion period to clarify any remaining points. Please read our comment above, addressed to all reviewers, first.
>
>
> > [W1.1] would be significantly improved with an architecture diagram of the policy and the classifier.
>
>
> While we would be happy to create a diagram that can help further clarify our method, we are unsure what exactly you are asking for, and would therefore appreciate some clarifications from your side.
>
>
> We are not introducing any new architecture and are directly using the standard Perceiver IO model without any changes to construct our policy and classifier (cf. S 3.2).
> Therefore, ‘an architecture diagram of the policy and the classifier’ would just be Figure 2 of the Perceiver IO paper (https://arxiv.org/abs/2107.14795) twice, once for the agent and once for the model. We are happy to refer to this figure in our paper for clarification, if you think this helps.
>
>
> Figure 1 clarifies the procedure of A2MT, which is one of our main conceptual contributions.
> Perhaps a more abstract version of Figure 1, such as https://snipboard.io/FNkU0n.jpg would be in the spirit of your comment?
>
>
> > [W1.2] S3.1: It is unclear if the randomly drawn starting value for the counter is drawn once or multiple times. The example sequence given: “2103210210” appears to be only realizable if it is drawn multiple times, assuming values: [2, 3, 2]. Please clarify this in the paper.
>
>
> You are correct, we draw multiple starting values and then concatenate the countdown sequences, i.e. the starting values `[2,3,2]` lead to the countdowns `[[2,1,0],[3,2,1,0],[2,1,0]` which are then concatenated to yield the `counter` modality `[2,1,0,3,2,1,0,2,1,0]`. Drawing multiple starting values ensures the agent needs to repeatedly attend to the counter modality. Apologies for the confusion, we have clarified this in our description of the synthetic dataset in S 3.1.
>
>
> > [W1.3] The experiment presented in Table 3 is unclear. A mask rate of 0 results in AudioSet mAP of 0.370, which appears to correspond to the 0.370 values in Table 2 and Table 1.  [...] My main guess here is that the Mask Rate in Table 3 is the rate of masking the test data, not the rate of masking the training data. [...]
>
>
> Sorry for all the mental gymnastics you had to go through, and thank you for drawing our attention to the importance of highlighting clearly the difference between masking at evaluation and training time! You are entirely correct with your conclusions:
>
>
> The masking strategies (M1-M3) are those applied during training only, and the masking probabilities for M1 and M3 are fixed throughout the experiments. We had detailed this in Appendix C.4 but have now added clarifications to 4.2 directly. We have also added a sentence stressing that (M1-M3) are only applied during training, and clarified the appropriate captions to stress when masking is applied during training (Table 2 (formerly 1)) and when during evaluation (Tables 3-4 (formerly 2-3)).
>
>
> > [W1.4] S5.1 would be improved by presenting the metrics in a table, rather than inline, in order to increase their visibility.
>
>
> Absolutely! That is a great suggestion, and we’ve implemented it in the revised draft.
>
>
> **Please read part 2 of our reply next.**

---

> > ### Author Response · Authors · 2023-05-05
> > **Author Response to Reviewer 5VUY (Part 2)**
> >
> > **Please read part 1 of our reply first.**
> >
> >
> > > [W1.5] The related work section could be improved by including discussion of the “anytime prediction” framework and approaches (e.g., but not limited to, [A,B,C,D] in the References section below),
> >
> >
> > Thanks for the suggestion of the anytime prediction literature as additional related work. We have had a look at the suggested references [A-D] and have included [A, B, D] as related work in our discussion.
> > [C] only mentions anytime prediction in passing. Could you perhaps elaborate on its relevance here?
> >
> >
> > >  in which a learner’s output can be computed at “any time” and improves with the addition of more inputs.
> >
> >
> > As far as we understand, anytime prediction is about improving predictions with additional _computation_ time, not with additional inputs. At least, we understood this to be the case for the references that you have provided. E.g. [A] writes “anytime algorithms are algorithms whose quality of results improves gradually as computation time increases”, or [C] writes “anytime predictors [...]
> > trade computation time for predictive accuracy by selecting from a set of simpler candidate predictors”.
> >
> >
> > We wish to highlight some crucial differences---to the best of our understanding---between anytime prediction and active feature acquisition below.
> >
> >
> > 1) Most importantly, anytime prediction implicitly assume that data is cheap (and thus fully observed) while computation is expensive. A2MT assumes the opposite: data acquisition is extremely expensive for each sample and thus justifies a significant amount of compute.
> >
> >
> > 2) A2MT assumes temporally causal data acquisition. In other words, our agents experience the _time_ dimension of the data, and it cannot go back (or forward) in time and acquire an observation for a past (or future timestep).  In anytime prediction, only the _time_ it takes to predict is counted, i.e.~the data (even if temporal) is usually assumed to be fully observed already.
> >
> >
> > 3) Our Perceiver IO based predictor can predict for arbitrary sequence length, i.e. at ‘any time in the sequence’. However, we believe this is different to the meaning of ‘any time’ in anytime prediction, which refers to controlling the computation time of the prediction. For example, SpeedBoost [C] has an additive structure, where additional cheap predictors can be added iteratively to improve the result. In contrast, with Perceiver IO, the predictions are neither iterative (if we want to predict for a longer sequence, we have to compute a separate complete forward pass) nor anytime (we always have to compute a full forward pass).
> >
> >
> > However, it is true that the computation cost reduces with sequence length for our Perceiver IO predictor. And thus, one has some rudimentary control over the computation cost. However, this is a feature of the Perceiver IO model and _not_ anything we introduce with A2MT.
> >
> >
> > It is probably easier to relate anytime prediction to ‘efficient video classification’ (which we cite already), which tries to reduce computation cost in video classification, often by subselecting informative frames from videos. For example AdaFrame (Wu, 2019b), which we cite already, spends compute in relation to the difficulty of classifying an input.
> >
> >
> > Again, thank you for these references. We have added them, and please let us know if you have any further comments.
> >
> > > [W2.1] The experiments in S5.1 and Fig A.1 could be improved with more metrics of counter acquisition [...]
> >
> >
> > *([edit]: See latest reply 'Additional Metrics for Synthetic Scenario' below. We have updated the paper draft to include these metrics.)*
> >
> >
> > These are all great suggestions! We will get to work right away to determine the (average) optimal number of acquisitions for the digit and counter modality for the synthetic scenario. As mentioned above, we wanted to prioritize responding quickly to get the discussion started early, so please understand that we cannot yet provide this result. We will update you with another message as soon as we have it available!
> >
> >
> > Furthermore, we will also aim to compute the ‘accuracy of the attention to the counter/digit’ variable that you suggest. We agree that these should be even more informative than a comparison to the ‘optimal rate’. We should have a copy of the raw prediction data for the agent on the synthetic scenario that allows us to compute these values.
> >
> >
> > Note that the minimum sequence length in our data generation is 2. This affects optimal behavior: the agent never needs to acquire the counter modality for the last timestep: If the countdown is incomplete, we can predict its value, and if a new counter starts, it no longer matters because the counter modality is larger than zero.
> >
> > While this might make it a little bit more challenging to compute the ‘accuracy of the attention’ values, we believe it should still be possible.
> >
> >
> > **Please read part 3 of our reply next.**

---

> > > ### Author Response · Authors · 2023-05-05
> > > **Author Response to Reviewer 5VUY (Part 3)**
> > >
> > > **Please read parts 1 and 2 of our reply first.**
> > >
> > > > [W2.2] The experiments in Table 4 and Table 5 would be significantly improved if they included prior work on these same classification tasks, both from approaches that are cost-insensitive (e.g. any prior classification approaches, which I’m fairly confident exist) and those that are cost-sensitive, if they exist on these dataset (I don’t know such exist on these datasets). [...]
> > >
> > > Cost-insensitive baselines: We believe there may be a misunderstanding here. Our predictive model $f$ (cf. S 4.2) is a completely standard Perceiver IO classification model, i.e. we believe it should classify as ‘any prior classification approaches’. We report its performance on fully and partially observed (masking at evaluation time) data in Tables 3 and 4 (previously 2 and 3). During A2MT reinforcement learning we train the agent, which influences input selection, but keep the predictive model fixed. For a fair comparison of our agent to a baseline, we think the random-rate and random-1hot make the most sense, where we ablate if our agent learns clever acquisition behavior that adapts to the information content in each sequence. Note that the random-rate and random-1hot ablations evaluate this pre-trained Perceiver IO model on subsets of the AudioSet/Kinetics input data. In this way, we believe our ablations can be viewed as a ‘cost-insensitive’ approach in the sense that they are not clever about how to spend their acquisition cost.
> > >
> > >
> > > Given that we include performance metrics for the Perceiver IO (which should classify as ‘any prior classification approach’ model under a variety of input configurations) we would appreciate some clarification for what exactly you are asking for.
> > >
> > >
> > > If you think it is useful, we can include numbers from other methods from the literature about performance values reached on Kinetics and AudioSet?  However, we do not think these would be too useful, as we ultimately care about evaluating the _acquisition strategy of our agent_, which is best compared  _relative_ to a fixed predictive model!
> > >
> > > Lastly, we note that many approaches in the literature may not be able to handle arbitrarily masked out modalities, which is a major reason that we decided to go with Perceiver IO in the first place.
> > >
> > >
> > > Cost-sensitive baselines: We are also not aware of any suitable baselines, which is why we frame the introduction of the A2MT scenario as part of our contribution!
> > >
> > >
> > > > It is unclear how the ranges of cost values investigated in Table 4 and Table 5 were chosen. [...]
> > >
> > >
> > > Thanks for bringing this to our attention. We agree that this should absolutely be discussed!
> > >
> > > We found valid cost ranges through experimentation on both AudioSet and Kinetics. As a rule of thumb, we aimed for acquisition costs such that $T$ times the cost is smaller than the observed predictive loss on fully unmasked data. If this were not the case, it is too attractive to not acquire any input samples. Note that predictive losses between AudioSet and Kinetics are significantly different; AudioSet is a multi-class (per instance) problem and Kinetics is not.
> > > As costs are directly traded off with predictive loss in our objective, cf. eq. 1, this explains the different cost magnitudes between the datasets.
> > >
> > >
> > > Once we found a valid cost value, we increased and decreased costs to find a range of costs that leads to varied agent behavior.
> > > First we increased costs until the agent (almost) did not make any acquisitions anymore. Tables 5 and 6 (previously 4 and 5) show this requires adjusting the costs across multiple magnitudes. Note that acquisition rates of about $0.05$ correspond to acquiring a single segment for our total of $T=25$ segments. As the agent already acquires only a single segment, additional cost increases are therefore not interesting from the perspective of A2MT. We also decreased costs until agent behavior became stagnant. For AudioSet at the lowest cost, the agent acquires (almost) all segments of the audio signal, so further decreasing the cost would not change behavior. For Kinetics, we have actually decreased the cost further, but this does not change agent behavior. In fact, agent acquisition rates are already constant for the three lowest costs in Table 6 (previously 5). In other words, the agent does not learn to acquire more than 20% of the image modality, at any cost, because it does not see a benefit for it for prediction. This fits with our observations in Table 4 (previously 3), where performance on Kinetics is almost identical until mask rates are increased above 80%, which points towards the repetitive nature of the image modality in the Kinetics dataset.
> > >
> > >
> > > We hope this clarifies how we chose mask values, that the reported cost ranges cover the majority of possible agent behavior and are hence sufficient, and why the ‘underperforming the ablations’ is unlikely to be a function of cost.
> > >
> > >
> > > Thanks again for raising this! We have added the discussion in Appendix C.

---

> > > > ### Author Response · Authors · 2023-05-10
> > > > **Author Response to Reviewer 5VUY (Additional Metrics for Synthetic Scenario)**
> > > >
> > > > Dear reviewer 5VUY,
> > > >
> > > >
> > > > Thanks again for your suggestion of comparing our agent's acquisition behavior to an oracle strategy for the synthetic scenario.
> > > >
> > > >
> > > > We have just uploaded a revised version of the paper that contains this evaluation. In the main paper (S 5.1 and Table 4) we have added a comparison to the optimal acquisition rates per modality. For more detailed insights, we have also computed the confusion matrices which we report in Appendix A (Table A.1 and A.2).
> > > > The results show that the agent largely follows the optimal acquisition strategy for the `digit` modality, and that there is significant room for improvement for the `counter` modality.
> > > >
> > > >
> > > > We hope that the revised version of the draft addresses your comment [W 2.1]!
> > > >
> > > >
> > > > We would further appreciate it if you could respond to our requests for clarification, in particular those regarding [W1.1, W2.2], as soon as possible.

---

> > > > > ### Author Response · Authors · 2023-05-22
> > > > > **Author Response to Reviewer 5VUY**
> > > > >
> > > > > Dear reviewer 5VUY,
> > > > >
> > > > >
> > > > > Thanks again for your thoughtful review, which we believe has already helped improve the paper!
> > > > >
> > > > >
> > > > > We would really appreciate it if you could respond to our requests for clarification, in particular regarding your points [W1.1] and [W2.2], while we can still make changes to the submission.
> > > > >
> > > > >
> > > > > Although we have already made significant effort to improve the paper based on your feedback, we would like to make sure that our answers fully address your concerns.

---

### Review · Reviewer_Xafx · 2023-05-02

**Summary Of Contributions:**

The paper has three contributions. First, it introduces a so-called A2MT task, which abstractly allows the agent to query temporal and multimodal data, e.g., video or audio stream. At each timestep, the agent can select some modalities (possibly none or all) and view their entry corresponding to the given timestep, e.g., a given frame in the video stream. The goal is to acquire enough data to predict some labels attached to the temporal and multimodal data (possibly some function of the data). Each query has its cost, possibly dependent on the modality, which creates a tug-of-war dynamics, where the agent wants to increase the chances of predicting the label, and at the same time, minimize the number of queries. The paper's second contribution is the actual instance of A2MT consisting of synthetic data (sort of a guessing numbers game) and the datasets based on AudioSet and Kinetics, introduced in other papers. Finally, the paper proposes a deep-learning system composed of a prediction model (here based on Perceiver IO architecture) and a policy, which makes a decision on which modality to query at each step. The paper includes experimental analysis, including positive and negative results.

**Audience:**

Yes

**Broader Impact Concerns:**

No concerns.

**Claims And Evidence:**

Yes

**Requested Changes:**

I like the paper: it sets an interesting problem, proposes a learning system that aims to solve the tasks, and discusses some interesting experimental phenomena. However, I would like the Authors to improve the clarity, the technical details, and discuss some of the alternative reasons why the policy does not learn adaptive behavior (for details, see the Weakness section). If the Authors decide to follow up with this, I am willing to set "Claims And Evidence" to "Yes".

**Strengths And Weaknesses:**

**Strengths:**
* The problem in the paper is ambitious, and the model chosen is a relatively new attention-based architecture that aims to decouple the input length from the model’s residual stream dimension.
* The problem is well-defined, and its relevance is described compellingly.
* The difficulty of datasets scales well from more simple to more complex.
* The paper provides some interesting observations, including a change from overfitting to generalization for the simple synthetic dataset, the comparison against random agents, the 'negative' result concerning adaptivity (see also Weakness paragraph), and the relationship between loss and entropy.

**Weaknesses:**
* The datasets should be clearly described in the main body of the paper. In several places, it is pretty hard to follow what are the shapes of the data or what part of the data the evaluations are performed on (e.g., Table 1-3).
* The paper's clarity would improve if the tables’ and figures’ captions were self-explanatory. One must find information scattered across the paper to understand better what is happening. For instance, in Table 4-5, it not entirely clear what 'Agent Acquisition Rate' is (it is hard to find a definition of that quantity, with a caption of Figure A.1 being some proxy of that), what are the Random baselines (page 8), why is only one cost per acquisition (page 7), what dataset is used for the calculations for Kinetics.
* The paper structures A2MT as a reinforcement learning problem by adding a temporal aspect, actions, sparse rewards, and using an RL algorithm for training (A2C). However, the exact setup is not clearly defined (typically, one describes the underlying MDP or a similar object). Some deviations from the standard setup include the reward being defined by a performance of another learned model or the policy depending on the whole trajectory (and sometimes more).
* The description of the synthetic dataset could be improved. For instance, “the counter modality repeatedly counts down from a randomly drawn staring value” is somewhat confusing. The lack of clear description here makes the analysis of Figure 4 hard to follow.
Describing Gumbel softmax as an RL algorithm is perhaps an abuse of notation.
* Random agents (baselines) are unclear: how to compute the acquisition rate for 'Random-Rate' agent, or how is the “fixed set of equidistantly spaced timesteps” chosen?
* The paper provides a 'negative' result by showing that the learned agent does not learn a policy that adapts to inputs. This is interesting (as mentioned in the Strengths paragraph); however, the justification of the Authors’ “there might not be enough signal from the model predictions for the agent to anticipate how acquisitions affect the reward” or “method struggles to learn acquisition strategies and representations simultaneously”. This leaves the reader slightly unsatisfied and unsure whether the Authors’ made enough effort to clarify the issue. Some of the reasons could include (1) Some specific features of the data (e.g., periodicity); (2) Too short learning; (3) Difficulty in learning a policy to match a pre-trained predictive model; (4) Distribution shift in masking from pre-training to policy training; (5) Not enough regularization (the Authors’ mention the model’s overfitting), etc.

---

> ### Author Response · Authors · 2023-05-05
> **Author Response to Reviewer Xafx (Part 1)**
>
> Dear reviewer Xafx,
>
>
> Thank you for your hard work and helpful feedback. We have very gladly incorporated many of your excellent suggestions! Please read our comment above, addressed to all reviewers, first.
>
>
> > The datasets should be clearly described in the main body of the paper. In several places, it is pretty hard to follow what are the shapes of the data or what part of the data the evaluations are performed on (e.g., Table 1-3).
>
>
> Thanks for pointing this out!
> We have added a discussion of input shapes in prominent places in section 5.1 and 5.2.
> We have further clarified the captions of Tables 2-4 (previously 1-3).
>
>
> If there are any other suggestions you have for clarifications, please let us know; we would be happy to implement them.
>
>
> > The paper's clarity would improve if the tables’ and figures’ captions were self-explanatory. One must find information scattered across the paper to understand better what is happening. For instance, in Table 4-5, it not entirely clear what 'Agent Acquisition Rate' is (it is hard to find a definition of that quantity, with a caption of Figure A.1 being some proxy of that), what are the Random baselines (page 8)
>
>
> To make Tables 5 and 6 (previously 4 and 5) more self-explanatory we have added a clarification of the 'Agent Acquisition Rate’ as well as the ‘Random baselines’.
>
>
> >  why is only one cost per acquisition (page 7),
>
>
> Sorry, we are not quite sure what you mean by ‘one cost per acquisition (page 7)’? In fact, the word ‘cost’ does not appear on page 7 at all, as far as we can tell.
>
>
> > what dataset is used for the calculations for Kinetics.
>
>
> We have clarified that, like for AudioSet, we are using the held out Kinetics _test_ set for our results in Table 6 (previously 5). Does this address your comment?
>
>
> > The paper structures A2MT as a reinforcement learning problem by adding a temporal aspect, actions, sparse rewards, and using an RL algorithm for training (A2C). However, the exact setup is not clearly defined (typically, one describes the underlying MDP or a similar object). Some deviations from the standard setup include the reward being defined by a performance of another learned model or the policy depending on the whole trajectory (and sometimes more).
>
>
> Thank you for this excellent suggestion. We agree that describing the MDP underlying A2MT would be an interesting contribution. We have added a section discussing A2MT from the perspective of a Partially Observable MDP in Appendix D, and we refer to it prominently in Section 2.
> (We originally planned to just add a short section to the main paper, however, it turns out that framing A2MT as POMDPs is rather complicated, and the resulting POMDP is unusual in terms of the state and action space construction. We would be happy to incorporate any further feedback on Appendix D from your side!)
>
>
> > The description of the synthetic dataset could be improved. For instance, “the counter modality repeatedly counts down from a randomly drawn staring value” is somewhat confusing. The lack of clear description here makes the analysis of Figure 4 hard to follow.
>
>
> We have extended and improved the description of the counter modality in Section 3, and extended the caption of Figure 4. Please let us know if you have any further suggestions.
>
>
> > Describing Gumbel softmax as an RL algorithm is perhaps an abuse of notation.
>
> Thanks for pointing us towards this. It was not our intention to construe the Gumbel-Softmax estimator as a general RL algorithm. We have clarified in S 4.1 that we are able to use the Gumbel-Softmax estimator only due to the “simple unmasking effect of actions in A2MT”.
>
>
> **Please see Part 2 of our reply next.**

---

> > ### Author Response · Authors · 2023-05-05
> > **Author Response to Reviewer Xafx (Part 2)**
> >
> > **Please read part 1 of our reply first.**
> >
> > > Random agents (baselines) are unclear: how to compute the acquisition rate for 'Random-Rate' agent, or how is the “fixed set of equidistantly spaced timesteps” chosen?
> >
> >
> > Apologies for the lack of clarity. We have extended our exposition in the `Random Ablations’ paragraph in S 5.5.2 and hope it is more clear now. We now write:
> >
> > “Therefore, we compare the performance of our agents against two ablations that we call 'random-rate' and 'random-1hot'.
> > For both, we first compute the average acquisition rate of the agent per modality, i.e. the average fraction of timesteps at which it acquires.
> > For the `random-rate` ablation, we acquire modalities with a fixed Bernoulli probability per timestep equal to the average acquisition rate of the agent.
> > Additionally, we construct the `random-1hot` ablation, by acquiring at a fixed number number of timesteps per modality that are equidistantly spread across the sequence.
> > The number of acquisitions is chosen such that we match the average number of agent acquisitions per modality.
> > (Usually the number of agent acquisitions is not an integer, and so we remove some acquisition probability for the last of the fixed timesteps.)
> > See fig. 5 for an illustration of the ablations.
> >
> >
> > These are ablations rather than baselines, as they use the per-modality acquisition rates found by the agent, and thus incur the same cost as the agent.
> > However, potentially unlike the agent, the ablations do not act adaptively: they acquire with the same fixed probabilities for each sequence.
> > If we find that our agent can consistently outperform both ablations, this is supporting evidence for adaptive behavior in the agent, adjusting its acquisitions to the information in each input.”
> >
> >
> > > discuss some of the alternative reasons why the policy does not learn adaptive behavior
> > > Some of the reasons could include (1) Some specific features of the data (e.g., periodicity); (2) Too short learning; (3) Difficulty in learning a policy to match a pre-trained predictive model; (4) Distribution shift in masking from pre-training to policy training; (5) Not enough regularization (the Authors’ mention the model’s overfitting), etc. [...]
> >
> > Thanks for these suggestions! We have extended our discussion in S 7 and also now include the reasons that you have suggested.
> >
> > We did not add reason (5), as overfitting was only a problem for the synthetic scenarios and not the more complex audio-visual datasets which we discuss in S 7.
> >
> >
> > Thanks again for you review and please let us know should there be anything else remaining that keeps you from changing “Claims and Evidence” to “Yes”.

---

> > > ### Comment · Reviewer_Xafx · 2023-05-15
> > > **Reviewer Response to the Authors**
> > >
> > > Dear Authors,
> > > Thank you for taking the time to address my concerns. I changed my score accordingly (as promised). I think that some points could still be improved in the camera-ready version:
> > > * In Tables 5-6 it could be helpful to mention that only one ("informative") modality has a cost, so only one cost is mentioned in the table.
> > > * Sections 3.2 and 5.2 have the same name and could be merged.
> > > * In the description of Random-1Hot agent, it is still unclear what the fixes set of equidistant steps are. For instance, in Fig 5c, why is there no acquisition at timestep 0, or how is the frequency computed for the last timestep (it is only mentioned that some probability is removed). Also, there appeared a typo 'number number' on page 9.

---

> > > > ### Author Response · Authors · 2023-05-15
> > > > **Author Response to Reviewer Xafx**
> > > >
> > > > Dear reviewer Xafx,
> > > >
> > > >
> > > > We are very happy to hear that our replies and revisions were able to address your concerns, and we thank you for increasing your score.
> > > >
> > > >
> > > > We will gladly implement the points you mention in the camera-ready version of the paper.
> > > >
> > > >
> > > > Lastly, we would like to thank you for engaging with us during the rebuttal period.

---

### Author Response · Authors · 2023-05-05
**Author Response to All Reviewers**

We thank all reviewers for their careful consideration, insightful comments, and helpful suggestions. We are glad that the paper was generally well received, and hope that our responses and paper updates will alleviate any concerns you raised. We believe your input has already helped improve the paper and look forward to engaging with you further during the discussion period.



We were glad to see all reviewers recognize our contribution of introducing the Active Acquisition for Multimodal Temporal Data (A2MT) scenario, highlighting the problem as ‘novel and in time’ (SDJ9), ‘interesting’ (Xafx, 5VUY), ‘ambitious’ (Xafx), and ‘well-defined’ (Xafx). We appreciate you felt A2MT is ‘compellingly’ (Xafx) relevant with a motivation that is ‘clear and evident’ (sDJ9). Further, we were glad to see you highlight our empirical evaluation as ‘thorough’ (sDJ9), showing ‘interesting experimental phenomena’ (Xafx), demonstrating ‘the potential of the proposed method for real-world applications’ (sDJ9), ‘of interest to the TMLR community’ (sDJ9), including ‘very important ablations’ (5VUY), and highlighting our synthetic scenario as a ‘good regime for further experimentation’ (5VUY).


**We have just uploaded a revised version of the paper draft, hopefully addressing many of your comments already.**


In order to avoid confusion, we note that we have added a new table to the draft, which is now table 1. Therefore, all table references in your reviews are now off by one. In our replies, we try to avoid any confusion by using a ‘Table X (previously X-1)’ notation.


Please see our individual replies to each of you below, where we detail the changes we have made in response to your comments or ask for clarifications whenever we were not sure how to best act on one of your remarks. We hope you will engage with us during the discussion period to clear up these uncertainties and allow us to improve the paper! Lastly, because we feel it is important to engage with you as soon as possible, we have not yet been able to address all comments in our revised draft. We will point out in our individual replies to you, whenever we are currently actively working on a solution.

---

### Author Response · Authors · 2023-05-11
**Author Response: Asking For Engagement Before the End of the Discussion Period**

Dear reviewers,


We would like to thank you again for your hard work. In response to your helpful comments we have submitted responses below and made considerable improvements to the paper draft.


**The discussion period will end Tuesday next week, and we would like to kindly request you to revisit our revised manuscript and the corresponding response to your review at your earliest convenience.**


We would be interested to hear if you feel we have adequately addressed your initial concerns. Most importantly, we felt it was necessary to ask clarifying questions on some of your comments, and we would really like to engage with you to fully clear these up!
In particular for reviewers 5VUY and sDJ9, there are significant open points of discussion.


We are committed to integrating your feedback and improving the submission, and hope to hear from you soon.

---

### Decision · Action_Editors · 2023-06-15

**Recommendation:** Accept with minor revision

**Comment:**

This paper is well-written and introducing an interesting problem. You have already addressed many of the reviewers concerns, so I won’t reiterate those here. The decision is Accept with Minor Revisions, where I (the action editor) can check that a few small things are implemented.

The reviewers largely seem happy with the changes. A few requests based on the reviewers reviews:
1. One reviewer asked for an architecture diagram of the policy and classifier. I suspect the reason for this request was that it is not clear how the classifier and the policy were jointly trained. Presumably, they were simply trained in parallel, with the policy producing the inputs and then that being handed to the classifier for training. I think this concern can be addressed by more clearly highlighting this training procedure, and stating what you said that you have two independent perceiver IO models. It would also be useful to explain why the Perceiver IO model easily handles missing data, since when searching in the original paper for the word “missing”, nothing seems to come up. A diagram of this might be useful way to explain this, but is not strictly necessary.

2. There was a nice request to formalize the MDP underlying this RL problem, and you did so in the appendix. However, this could use a bit of improvement. Its mostly ok except for the treatment of f (which makes this a nonstationary POMDP). Since f is not trained using RL, I wonder if it would be better to consider f as part of the MDP. In the simplest setting, f is fixed (somehow trained ahead of time) and the RL agent’s job is just to figure out what inputs to acquire to minimize cost while still getting good predictions. If f is changing, then this f could be part of the POMDP state (not visible to the agent).

3. One reviewer asked to include anytime prediction. I see how this is related, but quite different; its appreciated that you discussed it. However, the related work section is too sparse. Active perception and attention are large areas. For example, see a reasonably large survey in this work (I do apologize that it is my own, but of course, its the easiest one for me to think of, you can find other maybe more pertinent citations in it):
“Maximizing Information Gain in Partially Observable Environments via Prediction Rewards” Satsangi et al., AAMAS 2020

These should be very simple writing changes, thus the Accept with Minor Revisions.


**Audience:**

Yes

**Claims And Evidence:**

Yes

---

> ### Author Response · Authors · 2023-06-23
> **Author Response to Decision by Action Editors**
>
> Thank you for your hard work and helpful feedback. We are happy to see you recommend our submission be accepted after minor revision.
>
> We have gladly incorporated your three suggestions into the camera ready version of the submission, which we have just uploaded.
>
> 1. We have added additional details about the training procedure in Section 4.1, clearly highlighting that we have two independent Perceiver IO models and how joint training proceeds. We now also explain how/why Perceiver IO gracefully handles missing data in Section 4, which is referred to as 'masking' in the original Perceiver IO paper. Thanks for bringing this to our attention!
>
> 2. Thank you for your suggestions here. We agree it makes sense to not model the classifier $f$ as an action in the POMDP and have revised our formalization in Appendix D accordingly.
>
> 3. We have extended our discussion of related work in active perception and moved it to its own paragraph.